# Learning to design protein–protein interactions with enhanced generalization

**Anton Bushuiev**[1*]     **Roman Bushuiev**[1,4*]     **Petr Kouba**[1,2]     **Anatolii Filkin**[1]
**Marketa Gabrielova**[1]     **Michal Gabriel**[1]     **Jiri Sedlar**[1]     **Tomas Pluskal**[4]
**Jiri Damborsky**[2,3]     **Stanislav Mazurenko**[2,3]     **Josef Sivic**[1]

[1]Czech Institute of Informatics, Robotics and Cybernetics, Czech Technical University
[2]Loschmidt Laboratories, Department of Experimental Biology and RECETOX, Masaryk University
[3]International Clinical Research Center, St. Anne's University Hospital Brno
[4]Institute of Organic Chemistry and Biochemistry of the Czech Academy of Sciences

## Abstract

Discovering mutations enhancing protein–protein interactions (PPIs) is critical for advancing biomedical research and developing improved therapeutics. While machine learning approaches have substantially advanced the field, they often struggle to generalize beyond training data in practical scenarios. The contributions of this work are three-fold. First, we construct PPIRef, the largest and non-redundant dataset of 3D protein–protein interactions, enabling effective large-scale learning. Second, we leverage the PPIRef dataset to pre-train PPIformer, a new SE(3)-equivariant model generalizing across diverse protein-binder variants. We fine-tune PPIformer to predict effects of mutations on protein–protein interactions via a thermodynamically motivated adjustment of the pre-training loss function. Finally, we demonstrate the enhanced generalization of our new PPIformer approach by outperforming other state-of-the-art methods on new, non-leaking splits of standard labeled PPI mutational data and independent case studies optimizing a human antibody against SARS-CoV-2 and increasing the thrombolytic activity of staphylokinase.

## 1 Introduction

The goal of this work is to develop a reliable method for designing protein–protein interactions (PPIs). We focus on predicting binding affinity changes of protein complexes upon mutations. This problem, also referred to as $\Delta\Delta G$ prediction, is the central challenge of protein binder design (Marchand et al., 2022). The discovery of mutations increasing binding affinity unlocks application areas of tremendous importance, most notably in healthcare and biotechnology. Interactions between proteins play a crucial role in mechanisms of various diseases including cancer and neurodegenerative disorders (Lu et al., 2020; Ivanov et al., 2013). They also offer potential pathways for the action of protein-based therapeutics in addressing other medical conditions, such as stroke, which stands as a leading cause of disability and mortality worldwide (Feigin et al., 2022; Nikitin et al., 2022). Furthermore, the design of PPIs is also relevant to biotechnological applications, including the development of biosensors (Scheller et al., 2018; Langan et al., 2019).

While machine learning methods for designing protein–protein interactions have been developed for more than a decade, their generalization beyond training data is hindered by multiple challenges. First, training datasets for protein–protein interactions suffer from severe redundancy and biases. These inherent imperfections are difficult to identify and rectify using available algorithmic tools due to the low scalability of the latter. Second, the approaches employed for the train-test splitting of both large-scale unlabeled PPIs and small mutational libraries introduce data leakage, as interactions with near-duplicate 3D structures appear both in the training and test sets. As a result, performance estimates of machine learning models do not accurately reflect their real-world generalization capabilities. Third, commonly employed evaluation metrics do not fully capture practically important performance criteria. Finally, the design and training of the existing models for predicting

---

*These authors contributed equally.

binding affinity changes upon mutations are often prone to overfitting, as they do not fully capture the right granularity of the protein complex representation and the appropriate inductive biases.

In this work, we make a step towards more generalizable machine learning for the design of protein–protein interactions. First, we construct PPIRef[1], a comprehensive dataset of protein–protein interaction structures from the Protein Data Bank. Using iDist, a new scalable algorithm, we enhance the dataset quality by clustering PPIs and removing structural redundancy. Second, we introduce PPIFORMER[2,3], an $SE(3)$-equivariant transformer for modeling coarse-grained PPI structures. The transformer is pre-trained on PPIRef to generalize across diverse protein binders. Third, we fine-tune PPIFORMER to predict binding affinity changes upon PPI mutations ($\Delta\Delta G$), leveraging a thermodynamically motivated loss function. We demonstrate that PPIFORMER achieves state-of-the-art performance on multiple test sets, including non-leaking splits of standard mutational data, as well as case studies involving antibody design against SARS-CoV-2 and thrombolytic engineering.

## 2 RELATED WORK

**Predicting the effects of mutations on protein–protein interactions.**    The task of predicting the effects of mutations on protein–protein interactions measured by $\Delta\Delta G$ has been studied for more than a decade (Geng et al., 2019b). Traditionally, $\Delta\Delta G$ predictors relied on physics-based simulations and statistical potentials (Barlow et al., 2018; Xiong et al., 2017; Dehouck et al., 2013; Schymkowitz et al., 2005). In contrast, more recent machine learning approaches (Rodrigues et al., 2021; Pahari et al., 2020; Wang et al., 2020; Geng et al., 2019a) primarily rely on handcrafted descriptors of protein–protein interfaces. The latest generation of methods employs end-to-end deep learning on protein complexes, showing to surpass computationally intensive force field simulations in terms of predictive performance (Luo et al., 2023; Shan et al., 2022; Liu et al., 2021). In this study, we revisit this claim and show that the reported performance of the state-of-the-art methods may be overestimated due to leaks in the evaluation data and overfitting.

**Self-supervised learning for protein design.**    Self-supervised learning provides means to mitigate the high cost of collecting annotated protein mutation data for supervised learning. These methods, including protein language models (Meier et al., 2021), inverse folding (Hsu et al., 2022), and prediction of missing amino acid atoms (Shroff et al., 2019), leverage unannotated data to predict the effects of protein mutations. Recent advancements have also demonstrated that self-supervised pre-training on synthetic tasks enhances performance in various tasks related to protein structures (Zhang et al., 2023). Some of the latest binding $\Delta\Delta G$ predictors utilize pre-training from protein structures by learning to reconstruct native side-chain rotamer angles (Luo et al., 2023; Liu et al., 2021). Our work introduces a model pre-trained on 3D protein interaction structures aimed for the design of protein binders.

**Datasets of protein–protein interactions.**    Historically, datasets of protein–protein interactions have been falling under two categories: sets of hundreds of small curated task-specific examples (Jankauskaitė et al., 2019; Vreven et al., 2015), and larger collections of thousands of unannotated interactions, potentially containing biases and structural redundancy (Morehead et al., 2021; Evans et al., 2021; Townshend et al., 2019; Gainza et al., 2020). In this work, we construct a new dataset that is one order of magnitude larger than existing alternatives from the second category. We further refine our dataset by removing structurally near-duplicate

Table 1: Comparison of our PPIRef dataset with existing datasets of native protein complexes. The number of unique interfaces is estimated by deduplication using our iDist algorithm.

| Dataset | PPI structures | Unique interfaces |
|---|---|---|
| MaSIF-search | 6K | 5K |
| DIPS / DIPS-Plus | 40K | 9K |
| PPIRef (ours) | **322K** | **46K** (PPIRef50K) |

entries using our new scalable 3D structure-matching algorithm. This results in the largest available and non-redundant dataset of protein–protein interaction structures. Our algorithm for comparing

---

[1] https://github.com/anton-bushuiev/PPIRef
[2] https://github.com/anton-bushuiev/PPIformer
[3] https://huggingface.co/spaces/anton-bushuiev/PPIformer

protein–protein interactions contrasts with existing methods that rely on computationally expensive alignment procedures (Shin et al., 2023b; Mirabello & Wallner, 2018; Cheng et al., 2015; Gao & Skolnick, 2010a). Instead, we design our algorithm to enable large-scale retrieval of similar protein–protein interfaces by approximating their structural alignment. Our approach complements prior work on the efficient retrieval of similar protein sequences (Steinegger & Söding, 2017) and, more recently, monomeric protein structures (van Kempen et al., 2023).

## 3 PPIRef: New large dataset of protein–protein interactions

The Protein Data Bank (PDB) is a massive resource of over 200,000 experimentally obtained protein 3D structures (Berman et al., 2000). The space of protein–protein interactions in PDB is hypothesized to cover nearly all physically plausible interfaces in terms of geometric similarity (Gao & Skolnick, 2010b). Nevertheless, it comes at the expense of a heavy structural redundancy given by the highly modular anatomy of many proteins and their complexes (Draizen et al., 2022; Burra et al., 2009). To the best of our knowledge, there have been no attempts to quantitatively assess the redundancy of the large protein–protein interaction space represented in PDB and construct a balanced subset suitable for large-scale learning. We start this section by introducing the approximate iDist algorithm enabling fast detection of near duplicate protein–protein interfaces (Section 3.1). Using iDist, we assess the effective size and splits of existing PPI datasets (Section 3.2) and propose a new, largest and non-redundant PPI dataset, called PPIRef (Section 3.3).

### 3.1 iDist: new efficient approach for protein–protein interface deduplication

Existing algorithms to determine the similarity between protein–protein interfaces rely on structural alignment procedures (Gao & Skolnick, 2010a; Mirabello & Wallner, 2018; Shin et al., 2023a). However, since finding an optimal alignment between PPIs is computationally heavy, alignment-based approaches do not scale to large datasets. Therefore, we develop a reliable approximation of the well-established algorithm iAlign, which adapts the well-known TM-score to the domain of PPIs (Gao & Skolnick, 2010a). Our iDist algorithm calculates $SE(3)$-invariant vector representations of protein–protein interfaces via message passing between residues, which in turn enables efficient detection of near-duplicates using the Euclidean distance with a threshold estimated to approximate iAlign (Figure 4). The complete algorithm is described in Appendix A.

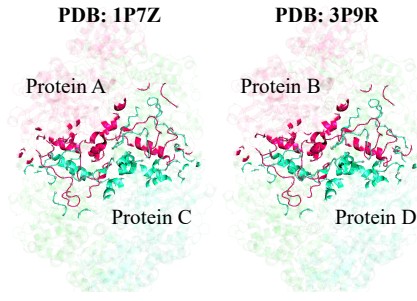

**PDB: 1P7Z**  **PDB: 3P9R**

Protein A  Protein B

Protein C  Protein D

Figure 1: An example of protein–protein interfaces from different folds of the DIPS dataset detected as near duplicates by our iDist method. Both PPIs come from the same KatE enzyme homooligomers with different single-point mutations. Notably, the symmetry of the complex itself yields 3 groups of 2 duplicates (from a single PDB entry with 6 PPIs). Furthermore, querying PDB with "KatE" results in 36 KatE complexes, yielding, therefore, 3 groups of 72 duplicates each. Our iDist approach can efficiently identify such structural near duplicates on a large scale.

To evaluate the performance of our iDist, we benchmark it against iAlign. We start by sampling 100 PDB codes from the DIPS dataset (Townshend et al., 2019) of protein–protein interactions and extract the corresponding 1,646 PPIs. Subsequently, we calculate all 2,709,316 pairwise similarities between these PPIs using both the exact iAlign structural alignment algorithm and our efficient iDist approximation. Employing 128 CPUs in parallel, iAlign computations took 2 hours, while iDist required 15 seconds, being around 480 times faster. Next, we estimate the quality of the approximation on the task of retrieving near-duplicate PPI interfaces. Using iAlign-defined ground-truth duplicates, iDist demonstrates a precision of 99% and a recall of 97%. These evaluation results confirm that iDist enables efficient structural deduplication of extensive PPI data within a reasonable timeframe, which is not achievable with other existing algorithms. The scalability of the method enabled us to analyze existing PPI datasets and their respective data splits used by the recent machine-learning approaches. The analysis is described below. Please refer to Appendix A for the additional details of the evaluation of iDist.

### 3.2 LIMITATIONS OF EXISTING PROTEIN–PROTEIN INTERACTION DATASETS

We apply iDist to assess the composition of DIPS – the state-of-the-art dataset of protein–protein interactions comprising approximately 40,000 entries (Morehead et al., 2021; Townshend et al., 2019). We construct a near-duplicate graph of DIPS by connecting two protein–protein interfaces if they are detected as near duplicates by iDist (see Figure 1 for an example). This results in a graph with 8.5K connected components, where the largest connected component comprises 36% of the interfaces. Notably, relaxing iDist duplicate detection threshold 1.5 times results in 84% of the interfaces forming a single component, indicating the high connectivity of the PPI space in DIPS. After iteratively deduplicating entries with at least one near duplicate, the dataset size drops to 22% of its initial size. These observations are in agreement with the hypothesis of Gao & Skolnick (2010b) suggesting high connectivity and redundancy of the PPI space in PDB.

Finally, we analyze DIPS data splits, estimating leakage by the ratio of test interactions with near duplicates in training or validation folds. We find that the split based on protein sequence similarity (not 3D structure as in our case) used for the validation of protein–protein docking models (Ketata et al., 2023; Ganea et al., 2021) has near duplicates in the training data for 53% of test examples, while the random split from Morehead et al. (2021) has 88% of test examples with a near duplicate in the training data. Figure 1 illustrates an example of such a leak in the original split (Ganea et al., 2021): the interface from the test fold (left) and a near-duplicate from the training fold (right). In Bushuiev et al. (2024), we demonstrate that high data leakage is present in most splits of PPI structures across multiple tasks due to the insufficiency of similarity measures commonly employed to test for data leakage and create the splits.

### 3.3 PPIREF: NEW LARGE DATASET OF PROTEIN–PROTEIN INTERACTIONS

We address the redundancy of existing PPI datasets by building a novel dataset of structurally distinct 3D protein–protein interfaces, which we call PPIRef. We start by exhaustively mining all 202,380 entries from the Protein Data Bank as of June 20, 2023. Subsequently, we extract all putative interactions by finding all pairs of protein chains that have a contact between heavy atoms in the range of at most 10Å. This procedure results in 837,241 hypothetical PPIs, further referred to as the raw PPIRef800K. Further, we apply the well-established criteria (Appendix A, Townshend et al. (2019)) to select only biophysically proper interactions. This filtering results in 322,454 protein–protein interactions comprising the vanilla version of PPIRef, which we name PPIRef300K. Finally, we iteratively filter out near-duplicate entries by applying the iDist algorithm. This deduplication results in the final non-redundant dataset comprising 45,553 PPIs, which we call PPIRef50K (or simply PPIRef). Table 1 illustrates that our dataset exceeds the total and effective sizes of the representative alternatives: DIPS (Townshend et al., 2019), DIPS-Plus (Morehead et al., 2021), and the dataset used to train the MaSIF-search model (Gainza et al., 2020), also used by the recent MaSIF-seed pipeline for *de novo* PPI design (Gainza et al., 2023).

## 4 LEARNING TO DESIGN PROTEIN–PROTEIN INTERACTIONS

Predicting how amino-acid substitutions affect binding energies of protein complexes ($\Delta\Delta G$) is crucial for designing protein interactions but is hindered by limited data, covering just a few hundred interactions (Jankauskaitė et al., 2019). This section presents our approach, including developing coarse-grained representations of protein complexes to avoid overfitting, introducing the $SE(3)$-equivariant PPIFORMER model to learn from the coarse-grained representations, and detailing our structural masked modeling for pre-training on the large PPIRef dataset. Finally, we present how we fine-tune PPIFORMER to predict $\Delta\Delta G$ values using a thermodynamically motivated loss function.

### 4.1 REPRESENTATION OF PROTEIN–PROTEIN COMPLEXES

In a living cell, proteins continually undergo thermal fluctuations and change their precise shapes. This fact is particularly manifested on protein surfaces, where the precise atomic structure is flexible due to interactions with other molecules. As a result, protein–protein interfaces may be highly flexible (Kastritis & Bonvin, 2013). Nevertheless, the available crystal structures from the Protein Data Bank only represent their rigid, energetically favorable states (Jin et al., 2023). Therefore, we

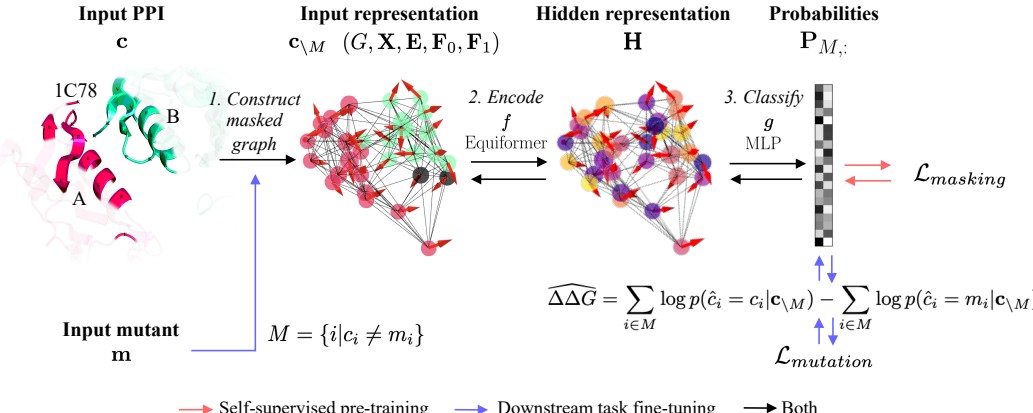

Figure 2: Overview of PPIFORMER. A single pre-training step starts with randomly sampling a protein–protein interaction $\mathbf{c}$ (in this example, the staphylokinase dimer A–B from the PDB entry 1C78) from PPIRef. Next, randomly selected residues $M$ are masked to obtain the masked interaction $\mathbf{c}_{\setminus M}$. After that, the interaction is converted into a graph representation $(G, \mathbf{X}, \mathbf{E}, \mathbf{F}_0, \mathbf{F}_1)$ with masked nodes $M$ (black circles). The model subsequently learns to classify the types of masked amino acids by acquiring $SE(3)$-invariant hidden representation $\mathbf{H}$ of the whole interface via the encoder $f$ and classifier $g$ (red arrows). On the downstream task of $\Delta\Delta G$ prediction, mutated amino acids are masked, and the probabilities of possible substitutions $\mathbf{P}_{M,:}$ are jointly inferred with the pre-trained model. Finally, the estimate $\widehat{\Delta\Delta G}$ is obtained using the predicted probabilities $p$ of the wild-type $c_i$ and the mutant $m_i$ amino acids via log odds (blue arrows).

aim to develop representations of protein complexes robust to atom fluctuations, as well as well-suited for modeling mutated interface variants. In this section, we define a coarse residue-level representation to allow for sufficient flexibility of the interfaces, which nevertheless captures the major aspects of the interaction patterns.

More specifically, consider a protein–protein complex (or interface) $\mathbf{c} \in \mathcal{A}^N$ of $N$ residues in the alphabet of amino acids $\mathcal{A} = \{1, \ldots, 20\}$. We consider the order of residues in $\mathbf{c}$ arbitrary, ensuring permutation invariance, a property critical for modeling the interfaces (Mirabello & Wallner, 2018). Next, we represent the complex $\mathbf{c}$ as a $k$-NN graph $G$, where the nodes represent the individual residues and edges are based on the proximity of corresponding alpha-carbon ($C_\alpha$) atoms. The graph is augmented by node-level and pair-wise features $\mathbf{X}, \mathbf{E}, \mathbf{F}_0, \mathbf{F}_1$. In detail, matrix $\mathbf{X} \in \mathbb{R}^{N \times 3}$ contains the coordinates of alpha-carbons of all residues. Next, all residue nodes are put into semantic relation by pair-wise binary edge features $\mathbf{E} \in \{0, 1\}^{N \times N}$ equal to 0 if residues come from the same protein partner and 1 otherwise. Finally, each node is associated with two kinds of features: type-0 $\mathbf{F}_0 \in \mathbb{R}^{N \times 20 \times 1}$, also referred to as scalars, and type-1 vectors $\mathbf{F}_1 \in \mathbb{R}^{N \times 1 \times 3}$. Features $\mathbf{F}_0$ capture the one-hot representations of wild-type amino acids $\mathbf{c}$, while vectors $\mathbf{F}_1$ are defined as virtual beta-carbon orientations calculated from the backbone geometry using ideal angle and bond length definitions (Dauparas et al., 2022).

Please note that $\mathbf{F}_0$ are invariant with respect to rototranslations and $\mathbf{F}_1$ are equivariant. Collectively, coordinates $\mathbf{X}$ and virtual-beta carbon directions $\mathbf{F}_1$ capture the complete geometry of the protein backbones involved in the complex. Our representation is agnostic to precise angles of side-chain rotamers, implicitly modeling their flexibility. The schematic illustration is provided in Figure 2.

## 4.2 PPIFORMER MODEL

In order to effectively learn from protein–protein complexes or interfaces $(G, \mathbf{X}, \mathbf{E}, \mathbf{F}_0, \mathbf{F}_1)$, i.e. respecting permutation invariance of amino acids and arbitrary coordinate systems of Protein Data Bank entries, we define PPIFORMER, an $SE(3)$-equivariant architecture. The model consists of an encoder $f$ and classifier $g$ such that $g(f(G, \mathbf{X}, \mathbf{E}, \mathbf{F}_0, \mathbf{F}_1))$ yields a probability matrix $\mathbf{P} \in [0, 1]^{N \times |\mathcal{A}|}$, where $P_{i,j}$ defines the probability of amino-acid type $j$ at residue $i$. Intuitively, matrix $\mathbf{P}$ represents amino acid probabilities in a protein complex based on its structure.

The core of our architecture is comprised of Equiformer $SE(3)$-equivariant graph attention blocks $f^{(l)}$ (Liao et al., 2023; Liao & Smidt, 2022). Each of the $L$ blocks (or layers) updates equivariant features of different types associated with all amino acids via message passing with an equivariant attention mechanism with $K$ heads. In detail, the input to the $l^\text{th}$ block is an original graph $G$ with coordinates $\mathbf{X}$ and edge features $\mathbf{E}$ along with a set of node feature matrices $\mathbf{H}_k^{(l)} \in \mathbb{R}^{N \times d_k \times (2k+1)}$ of different equivariance types $k \geq 0$ from the previous layer. Here, $d_k \geq 0$ is a hyper-parameter defining the number of type-$k$ hidden features, which is shared across all blocks, and $2k+1$ is their corresponding theoretical dimension. The input node features for the first layer are set to $\mathbf{F}_0, \mathbf{F}_1$. The output of each block is given by the updated node features of different equivariance types. Internally, all blocks lift hidden features up to the equivariant representations of the degree $\deg \geq 0$ which is an additional hyper-parameter. Taking the type-0 outputs of the final layer leads to invariant amino acid embeddings $\mathbf{H}$ as:

$$\mathbf{H}_0^{(1)}, \mathbf{H}_1^{(1)}, \ldots, \mathbf{H}_{deg}^{(1)} = f^{(0)}(G, \mathbf{X}, \mathbf{E}, \mathbf{F}_0, \mathbf{F}_1), \tag{1}$$

$$\mathbf{H}_0^{(l+1)}, \mathbf{H}_1^{(l+1)}, \ldots, \mathbf{H}_{deg}^{(l+1)} = f^{(l)}(G, \mathbf{X}, \mathbf{E}, \mathbf{H}_0^{(l)}, \mathbf{H}_1^{(l)}, \ldots, \mathbf{H}_{deg}^{(l)}), \tag{2}$$

$$\mathbf{H} := \mathbf{H}_0^{(L-1)}. \tag{3}$$

Collectively, we term the composition of transformer blocks $f^{(l)}$ (for $l \in \{0, \ldots, L-1\}$) as the encoder

$$f : G, \mathbf{X}, \mathbf{E}, \mathbf{F}_0, \mathbf{F}_1 \mapsto \mathbf{H} \tag{4}$$

with the property of $SE(3)$-invariance for any rotation $\mathbf{R} \in SO(3)$ and translation $\mathbf{t} \in \mathbb{R}^3$:

$$f(G, \mathbf{X}, \mathbf{E}, \mathbf{F}_0, \mathbf{F}_1) = f(G, \mathbf{X}\mathbf{R} + \mathbf{1}\mathbf{t}^T, \mathbf{E}, \mathbf{F}_0, \mathbf{F}_1\mathbf{R}). \tag{5}$$

To further estimate the probabilities of masked amino acids discussed below, we apply a 1-layer classifier with the softmax activation $g : \mathbb{R}^{N \times d_0 \times 1} \to [0, 1]^{N \times |\mathcal{A}|}$ on top of node embeddings $\mathbf{H}$ to obtain the probability matrix $\mathbf{P}$. The illustration of the model is provided in Figure 2 with the described pipeline depicted with black arrows.

### 4.3 3D EQUIVARIANT SELF-SUPERVISED PRE-TRAINING FROM UNLABELED PROTEIN−PROTEIN INTERACTIONS

In this section, we describe how we leverage a large amount of unlabeled protein–protein interfaces from PPIRef to train PPIFORMER to capture the space of native PPIs via masked modeling.

**Structural masking of protein–protein interfaces.** The paradigm of masked modeling has proven to be an effective way of pre-training from protein sequences (Lin et al., 2023). Nevertheless, while the masking of amino acids in a protein sequence is straightforward by introducing a special token, the masking of structural fragments is not obvious. Here, we leverage our flexible coarse-grained protein–protein complex representation to define masking by a simple change in the feature representation.

Having a protein complex or interface $\mathbf{c} \in \mathcal{A}^N$ containing $N$ amino acids from vocabulary $\mathcal{A}$ and a mask $M \subset \{1, \ldots, N\}$, we define the masked structure $\mathbf{c}_{\backslash M} \in (\mathcal{A} \cup \{0\})^N$ by setting amino acid classes at all masked positions $M$ to zeros. Consequently, when constructing one-hot scalar features $\mathbf{F}_0$ from $\mathbf{c}_{\backslash M}$, we set the corresponding rows to zeroes. Note that vector features $\mathbf{F}_1$ do not require masking since they do not contain any information about the types of amino acids, benefiting from using virtual beta-carbons instead of real ones. Additionally, the glycine amino acid, which lacks the beta-carbon atom, does not need special handling.

**Loss for masked modeling of protein–protein interfaces.** Having a protein–protein interaction $\mathbf{c} \in \mathcal{A}^N$ and a random mask $M \subset \{1, \ldots, N\}$, we use a traditional cross-entropy loss for training the model to predict native amino acids in a masked version $\mathbf{c}_{\backslash M} \in (\mathcal{A} \cup \{0\})^N$ of the native interaction. To additionally increase the generalization capabilities of PPIFORMER to capture potentially unseen or mutated interfaces, we further employ two regularization measures. First, we apply label smoothing (Szegedy et al., 2016) to force the model not to be overly confident in native amino acids, so that it can be more flexible towards unseen variants. Second, we weight the loss

inversely to the prior distribution of amino acids in protein–protein interfaces to remove the bias towards overrepresented amino acids such as leucine. The overall pre-training loss is defined as

$$\mathcal{L}_{masking} = -\sum_{i \in M} w_{c_i} \left[ (1 - \epsilon) \cdot \log p(\hat{c}_i = c_i | \mathbf{c}_{\setminus M}) + \epsilon \sum_{j \in \mathcal{A} \setminus c_i} \log p(\hat{c}_i = c_j | \mathbf{c}_{\setminus M}) \cdot \frac{1}{|\mathcal{A}|} \right]. \quad (6)$$

Here, $p(\hat{c}_i = c_i | \mathbf{c}_{\setminus M}) := P_{i,c_i}$ is the probability of predicting the native amino acid type $c_i$ of a masked residue $i \in M$. Next, the sum over all other, non-native, amino acid types $j \in \mathcal{A} \setminus c_i$ is the label smoothing regularization term where $0 < \epsilon < 1$ is the smoothing hyper-parameter. Further, $w_{c_i}$ is the weighting factor corresponding to the native amino acid $c_i$. Finally, the sum over $i \in M$ corresponds to the loss being evaluated over all masked residues.

### 4.4 Transfer learning for predicting the effects of mutations on protein–protein interactions

Predicting the effects of mutations on binding affinity ($\Delta\Delta G$) is a central task in designing protein–protein interactions. Nevertheless, collecting $\Delta\Delta G$ annotations is expensive and time-consuming. As a result, the labeled mutational data for binding affinity changes are scarce, not exceeding several thousand annotations (Jankauskaitė et al., 2019). Therefore, in our work, we aim to leverage the pre-trained PPIFORMER model for scoring mutations with minimal supervision.

**Thermodynamic motivation.** From the thermodynamic perspective, binding energy change $\Delta\Delta G$ (or alternatively denoted as $\Delta\Delta G_{wt \to mut}$) can be decomposed as follows:

$$\Delta\Delta G = \Delta G_{mut} - \Delta G_{wt} = RT(\log(K_{wt}) - \log(K_{mut})), \quad (7)$$

where $\Delta G_{mut}$ and $\Delta G_{wt}$ are binding energies of mutated and wild-type complexes, respectively (Kastritis & Bonvin, 2013), $R, T > 0$ are the gas constant and temperature of the environment, respectively, and $K_{wt}$ and $K_{mut}$ denote the equilibrium constants of protein–protein interactions, i.e. the ratios of the concentration of the complexes formed when proteins interact to the concentrations of the non-interacting proteins. The form of $\Delta\Delta G$ (Equation (7)) introduces symmetries into the problem of estimating the quantity. For example, predicting the effect of a reversed mutation $\Delta\Delta G_{mut \to wt}$ should satisfy the antisymmetry property $\Delta\Delta G_{wt \to mut} = -\Delta\Delta G_{mut \to wt}$. Available machine learning predictors either ignore the symmetry (Liu et al., 2021) or require two forward passes to estimate the quantity twice, for both directions, and combine the predictions to enforce the antisymmetry as $(\Delta\Delta G_{wt \to mut} - (-\Delta\Delta G_{mut \to wt}))/2$ (Luo et al., 2023).

**Predicting the effects of mutations on binding energy via the log odds ratio.** Here, in line with physics-informed machine learning (Karniadakis et al., 2021), we leverage the thermodynamic interpretation of $\Delta\Delta G$ to adapt the pre-training cross-entropy for the downstream $\Delta\Delta G$ fine-tuning. Having a complex $\mathbf{c} \in \mathcal{A}^N$ and its mutant $\mathbf{m} \in \mathcal{A}^N$, with the substitutions of residues $M \subset \{1, \dots, N\}$ such that $c_i \neq m_i$ for all $i \in M$, we estimate its binding energy change as:

$$\widehat{\Delta\Delta G} = \sum_{i \in M} \log p(\hat{c}_i = c_i | \mathbf{c}_{\setminus M}) - \sum_{i \in M} \log p(\hat{c}_i = m_i | \mathbf{c}_{\setminus M}), \quad (8)$$

where the $p$ terms are PPIFORMER output probabilities. The $\widehat{\Delta\Delta G}$ prediction is the log odds ratio, used by Meier et al. (2021) for zero-shot predictions on protein sequences. Intuitively, the predicted binding energy change upon mutation is negative (increased affinity) if the predicted likelihood of the mutated structure is higher than the likelihood of the native structure. When simultaneously decomposing Equation (7) and Equation (8), we observe that $\log(K_{wt})$ is estimated as $\sum_{i \in M} \log p(\hat{c}_i = c_i | \mathbf{c}_{\setminus M})$, and $\log(K_{mut})$ is estimated as $\sum_{i \in M} \log p(\hat{c}_i = m_i | \mathbf{c}_{\setminus M})$. Considering that the estimate of the wild-type likelihood is identical to the pre-training loss (Equation (6)) up to the regularizations, we reason that during pre-training PPIFORMER learns the correlates of $\Delta G$ values, whereas during fine-tuning it refines them to $\Delta\Delta G$ predictions. For fine-tuning through Equation (8), we use the MSE loss denoted as $\mathcal{L}_{mutation}$.

## 5 Experiments

In this section, we describe our protocol for benchmarking the generalization on the task of $\Delta\Delta G$ prediction and present our results. We begin by introducing the evaluation datasets, metrics, and

baseline methods (Section 5.1). Next, we show that our approach outperforms state-of-the-art machine learning methods in designing protein–protein interactions distinct from the training data (Section 5.2). Additionally, we show the benefits of our new PPIRef dataset and key PPIFORMER components through ablation studies in Appendix D.

## 5.1 EVALUATION PROTOCOL

**Datasets.** To fine-tune PPIFORMER for $\Delta\Delta G$ prediction we use the largest available labeled dataset, SKEMPI v2.0, containing 7085 mutations (Jankauskaitė et al., 2019). Prior works (Luo et al., 2023; Liu et al., 2021) primarily validate the performance of models on PDB-disjoint splits of the dataset. However, we find such approach not appropriate to measure the generalization capacity of predictors due to a high ratio of leakages. Therefore, we construct a new, non-leaking cross-validation split and set aside 5 PPI outliers to obtain 5 additional distinct test folds. Further, to simulate practical protein–protein interaction design scenarios, we perform additional evaluation on two independent case studies. These case studies assess the capability of models to retrieve mutations optimizing the human P36-5D2 antibody against SARS-CoV-2 and increasing the thrombolytic activity of staphylokinase. Please refer to Appendix C.1 for details.

**Metrics.** To evaluate the capabilities of models in prioritizing favorable mutations, we use the Spearman correlation coefficient between the predicted and ground-truth $\Delta\Delta G$ values. To evaluate the performance in detecting stabilizing mutations, we calculate precision and recall with respect to negative $\Delta\Delta G$ values. We also report additional metrics to enable comparison with prior work (Luo et al., 2023). However, in Appendix C.2, we emphasize that these metrics can be misleading when selecting a model for a practical application. For the independent retrieval case studies, we report the predicted ranks for all favorable mutations and evaluate the retrieval performance using the precision at $k$ metrics (P@$k$). We provide the details in Appendix C.2.

**Baselines.** We evaluate the performance of PPIFORMER against the state-of-the-art representatives in 5 main categories of available methods. Specifically, the flex ddG (Barlow et al., 2018) Rosetta-based (Leman et al., 2020) protocol is the most advanced traditional force-field simulator. GEMME is a state-of-the-art non-learning method based on evolutionary trees of sequences (Laine et al., 2019). The baseline machine learning methods are as follows: supervised RDE-Network (Luo et al., 2023), pre-trained on unlabeled protein structures and fine-tuned for $\Delta\Delta G$ prediction on SKEMPI v2.0; ESM-IF (Hsu et al., 2022), an unsupervised predictor of $\Delta\Delta G$ trained for inverse folding on experimental and AlphaFold2 (Jumper et al., 2021) structures; and MSA Transformer (Rao et al., 2021), an evolutionary baseline. Please see Appendix C.3 for details.

## 5.2 COMPARISON WITH THE STATE OF THE ART

**Prediction of $\Delta\Delta G$ on held-out test cases.** As shown in Table 2, on 5 challenging test PPIs from SKEMPI v2.0, PPIFORMER confidently outperforms all machine-learning baselines in 6 out of 7 evaluation metrics, being the second-best in terms of recall. We achieve a 183% relative improvement in mutation ranking compared to the state-of-the-art supervised RDE-Network, as measured by Spearman correlation. Importantly, this non-leaking evaluation reveals that traditional force field simulators, represented by state-of-the-art flex ddG, still outperform machine learning methods in terms of predictive performance. However, in terms of speed, they may not be applicable in typical real-world mutational screenings, being 5 orders of magnitude slower (see Appendix C.3).

Table 2: Test set performance averaged across five held-out protein–protein interactions selected from SKEMPI v2.0 for benchmarking (see Appendix C.1 for details). The standard deviation for PPIFORMER is estimated from three fine-tuning experiments with different random seeds.

| Category | Method | Spearman ↑ | Pearson ↑ | Precision ↑ | Recall ↑ | ROC AUC ↑ | MAE ↓ | RMSE ↓ |
|---|---|---|---|---|---|---|---|---|
| Force field simulations | FLEX DDG | 0.55 | 0.57 | 0.63 | 0.62 | 0.84 | 1.59 | 2.00 |
| Machine learning | GEMME | 0.38 | 0.41 | **0.60** | 0.49 | 0.74 | 2.16 | 2.81 |
| | MSA TRANSFORMER | 0.31 | 0.36 | 0.51 | 0.38 | 0.70 | 6.13 | 6.93 |
| | ESM-IF | 0.18 | 0.18 | 0.33 | 0.41 | 0.68 | 1.87 | 2.15 |
| | RDE-NET. | 0.24 | 0.30 | 0.54 | **0.65** | 0.67 | 1.70 | 2.02 |
| | PPIFORMER (OURS) | **0.44** ± 0.03 | **0.46** ± 0.03 | **0.60** ± 0.014 | 0.61 ± 0.012 | **0.78** ± 0.019 | **1.64** ± 0.011 | **1.94** ± 0.006 |

**Optimization of a human antibody against SARS-CoV-2.** Within a pool of 494 candidate single-point mutations of a human antibody, our model detects 2 out of 5 annotated mutations (using the top-10% threshold as defined by Luo et al. (2023)) that are known to be effective against SARS-CoV-2 (Table 3). The best among the other methods detect 3 out of 5 mutations. However, in contrast to the other methods, PPIFORMER successfully assigns the best rank to one of the 5 favorable mutations (P@1 = 100%). Besides that, PPIFORMER achieves superior performance when considering the ranks of all 5 mutations collectively, with the maximum rank for a favorable mutation not exceeding 21.46% and the mean rank of 11.60% (compared to the second best values of 51.42% and 18.26%, respectively). Overall, PPIFORMER is superior in prioritizing favorable mutations among random ones but does not prioritize 5 annotated mutants as high as the DDGPred and RDE-Network models. This suggests that there may be other, potentially more favorable, candidates in the unannotated pool of other 489 out of 494 single-point mutations.

Table 3: The performance of PPIFORMER and eight competing methods in retrieving 5 human antibody mutations effective against SARS-CoV-2. The middle section shows ranks (in percent) for individual mutations with predictions scored in the top 10% in bold, as in (Luo et al., 2023). The right section presents precision metrics. The values for baseline methods except for MSA Transformer and ESM-IF are reproduced from (Luo et al., 2023).

| Method | TH31W ↓ | AH53F ↓ | NH57L ↓ | RH103M ↓ | LH104F ↓ | P@1 ↑ | P@5% ↑ | P@10% ↑ |
|---|---|---|---|---|---|---|---|---|
| MSA TRANSFORMER | 56.88 | 42.11 | 63.56 | 49.19 | 18.83 | 0.00 | 0.00 | 0.00 |
| ESM-IF | 49.39 | 17.61 | 17.00 | 51.42 | 48.58 | 0.00 | 0.00 | 0.00 |
| ROSETTA | 10.73 | 76.72 | 93.93 | 11.34 | 27.94 | 0.00 | 0.00 | 0.00 |
| FOLDX | 13.56 | **6.88** | **5.67** | 16.60 | 66.19 | 0.00 | 0.00 | 4.08 |
| DDGPRED | 68.22 | **2.63** | 12.35 | **8.30** | **8.50** | 0.00 | 4.00 | **6.12** |
| END-TO-END | 29.96 | **2.02** | 14.17 | 52.43 | 17.21 | 0.00 | 4.00 | 2.04 |
| MIF-NET. | 24.49 | **4.05** | **6.48** | 80.36 | 36.23 | 0.00 | 4.00 | 4.08 |
| RDE-NET. | **1.62** | **2.02** | 20.65 | 61.54 | **5.47** | 0.00 | **8.00** | **6.12** |
| PPIFORMER (OURS) | 18.02 | **0.20** | **7.69** | 21.46 | 10.9 | **100** | 4.00 | 4.08 |

**Engineering staphylokinase for enhanced thrombolytic activity.** Within a pool of 80 annotated mutations, PPIFORMER precisely prioritizes 2 out of 6 strongly favorable ones (i.e., increasing the thrombolytic activity at least twice) as the top-2 mutation candidates (Table 4). In contrast, the second-best method, RDE-Network, identifies one mutation and assigns it a worse ranking (top-4). MSA Transformer provides accurate top-1 prediction but strongly loses performance on higher cutoffs and detecting two-fold activity-increasing mutations.

Table 4: The performance of PPIFORMER and three competing methods in retrieving the mutations on the staphylokinase interface that increase its trombolytic activity via enhanced affinity to plasmin. Here, 20 of 80 mutations are favorable for activity (right section), and 6 of them lead to at least two-fold enhancement (middle section). Similarly to Table 3, the rank values in bold are those in the top-10% predictions.

| Method | Mutations with ≥ 2× activity enhancement | | | | | | Activity enhancement | | |
|---|---|---|---|---|---|---|---|---|---|
| | | | KC135R | | KC74Q KC130E | KC74R KC130T | | | |
| | KC130A ↓ | KC130T ↓ | KC130T ↓ | KC135A ↓ | KC135R ↓ | KC135R ↓ | P@1 ↑ | P@5% ↑ | P@10% ↑ |
| MSA TRANSFORMER | 52.50 | 32.50 | 55.00 | 40.00 | 70.00 | 78.75 | **100** | 50.00 | 37.50 |
| ESM-IF | 45.00 | 33.75 | 46.25 | 25.00 | 42.50 | 58.75 | 0.00 | 0.00 | 25.00 |
| RDE-NET. | 51.25 | 33.75 | 22.50 | 15.00 | 27.50 | **5.00** | 0.00 | 50.00 | 62.50 |
| PPIFORMER (OURS) | 66.25 | 15.00 | **2.50** | 52.50 | 33.75 | **1.25** | **100** | **75.00** | **87.50** |

# 6 CONCLUSION

In this work, we have constructed PPIRef – the largest and non-redundant dataset for self-supervised learning from protein–protein interactions. Using this data, we have pre-trained a new model, PPIFORMER, designed to capture diverse protein binding modes. Subsequently, we fine-tuned the model for the target task of discovering mutations enhancing protein–protein interactions. We have shown that our model effectively generalizes to unseen PPIs, outperforming other state-of-the-art machine learning methods. This work opens up the possibility of training large-scale foundation models for protein–protein interactions.

ACKNOWLEDGMENTS

This work was supported by the Ministry of Education, Youth and Sports of the Czech Republic through projects e-INFRA CZ [ID:90254], ELIXIR [LM2023055], CETOCOEN Excellence CZ.02.1.01/0.0/0.0/17_043/0009632, ESFRI RECETOX RI LM2023069. This work was also supported by the European Union (ERC project FRONTIER no. 101097822) and the CETOCOEN EXCELLENCE Teaming project supported from the European Union's Horizon 2020 research and innovation programme under grant agreement No 857560. This work was also supported by the Czech Science Foundation (GA CR) through grant 21-11563M and through the European Union's Horizon 2020 research and innovation programme under Marie Skłodowska-Curie grant agreement No. 891397. Views and opinions expressed are however those of the author(s) only and do not necessarily reflect those of the European Union or the European Research Council. Neither the European Union nor the granting authority can be held responsible for them. Petr Kouba is a holder of the Brno Ph.D. Talent scholarship funded by the Brno City Municipality and the JCMM. We thank David Lacko for preparing the dataset of staphylokinase mutations.

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

APPENDIX

The first section of the appendix provides additional details on the evaluation of the iDist algorithm by benchmarking it against the iAlign structural alignment method, as well as the details of the PPIRef dataset (Appendix A). Next, we provide the implementation (Appendix B) and experimental (Appendix C) details, including the selection of training hyper-parameters, the construction of the mutation datasets, and the description of the evaluation metrics and baselines. Next, we discuss several ablation studies illustrating the importance of the proposed PPIFORMER components (Appendix D). Finally, we provide the details of the comparison of our method with the state-of-the-art $\Delta\Delta G$ predictors on the SKEMPI v2.0 test set (Appendix E).

## A DETAILS OF THE IDIST ALGORITHM AND PPIREF DATASET

In this section, we discuss the details relevant to the analysis and construction of large unannoted PPI datasets. First, we provide the complete description of the iDist deduplication algorithm and the calibration of its distance threshold. Second, we specify the construction criteria for our PPIRef.

---

**Algorithm 1** iDistEMBED

1: **Input:** Protein–protein interface $\mathcal{I}$ of $N$ residues from $C$ chains.
2: **Output:** Vector representation of the interface $\mathbf{z}_{\mathcal{I}}$.
3: // Get coordinates, features, and partner information
4: $\mathbf{X} \in \mathbb{R}^{N \times 3}, \mathbf{F} \in \mathbb{R}^{N \times d}, \mathbf{p} \in \{1, \ldots, C\}^N \leftarrow get\_residues(\mathcal{I})$
5: // Embed residues
6: **for** $i \leftarrow 1$ to $N$ **do**
7: $\quad J_{intra} \leftarrow \{j \in \{1, \ldots, N\} \mid p_i = p_j\}$
8: $\quad J_{inter} \leftarrow \{j \in \{1, \ldots, N\} \mid p_i \neq p_j\}$
9: $\quad \mathbf{m}_{intra} \leftarrow \frac{1}{|J_{intra}|} \sum_{j \in J_{intra}} \mathbf{f}_j \cdot e^{-\frac{\|\mathbf{x}_i - \mathbf{x}_j\|_2^2}{\alpha}}$
10: $\quad \mathbf{m}_{inter} \leftarrow \frac{1}{|J_{inter}|} \sum_{j \in J_{inter}} \mathbf{f}_j \cdot e^{-\frac{\|\mathbf{x}_i - \mathbf{x}_j\|_2^2}{\alpha}}$
11: $\quad \mathbf{h}_i \leftarrow \frac{1}{2}\mathbf{f}_i + \frac{1}{4}\mathbf{m}_{intra} - \frac{1}{4}\mathbf{m}_{inter}$
12: **end for**
13: // Embed interface
14: **for** $c \leftarrow 1$ to $C$ **do**
15: $\quad J_c \leftarrow \{j \in \{1, \ldots, N\} \mid p_j = c\}$
16: **end for**
17: $\mathbf{z}_{\mathcal{I}} \leftarrow \frac{1}{|C|} \sum_{c=1}^{C} \frac{1}{|J_c|} \sum_{j \in J_c} \mathbf{h}_j$
18: **return** $\mathbf{z}_{\mathcal{I}}$

---

**Algorithm 2** iDist

1: **Input:** Two protein–protein interfaces $\mathcal{I}$ and $\mathcal{J}$.
2: **Output:** Distance $\geq 0$.
3: $\mathbf{z}_{\mathcal{I}} \leftarrow$ iDistEMBED($\mathcal{I}$)
4: $\mathbf{z}_{\mathcal{J}} \leftarrow$ iDistEMBED($\mathcal{J}$)
5: **return** $\|\mathbf{z}_{\mathcal{I}} - \mathbf{z}_{\mathcal{J}}\|_2$

---

**iDist algorithm.** Algorithms 1 and 2 outline iDist, a fast method for detecting near-duplicate protein–protein interfaces. Algorithm 1 details the conversion of a protein–protein interface $\mathcal{I}$ into a vector representation $\mathbf{z}_{\mathcal{I}}$. In the first step (line 4), the features of the interface $\mathbf{X}$, $\mathbf{F}$ and $\mathbf{p}$ are extracted. The residue coordinates $\mathbf{X}$ are determined by the positions of the $C_\alpha$ atoms. Next, the residue vector features $\mathbf{F}$ are initialized with simple 20-dimensional one-hot encodings of amino acids. We have also experimented with ESM-1b features (Rives et al., 2021) but obtained slightly lower performance. We observe that the reduced performance could be attributed to ESM-1b bias-

ing the comparison towards entire protein chains rather than the interfaces. Finally, each residue is associated with a label indicating the chain it belongs to, forming the vector $\mathbf{p}$.

In the following steps (lines 5-12), depicted in Figure 3, the hidden representation $\mathbf{h}_i$ for each residue $i$ is constructed. A residue $i$ receives messages from all other residues in both the same chain ($J_{intra}$) and the partnering chains ($J_{inter}$), represented by exponential radial basis functions with $\alpha$ set to 16. The messages are then averaged to create contact patterns $\mathbf{m}_{intra}$ and $\mathbf{m}_{inter}$. The final representation $\mathbf{h}_i$ is obtained by averaging the difference $\mathbf{m}_{intra} - \mathbf{m}_{inter}$, followed by averaging with the features $\mathbf{f}_i$. The difference heuristic is inspired by the complementarity nature of many non-covalent bonds governing PPIs. When amino acids in both the intra- and inter-contexts of a residue are similar (i.e. not small–bulky or negatively–positively charged), the message $\mathbf{m}_{intra} - \mathbf{m}_{inter}$ is low.

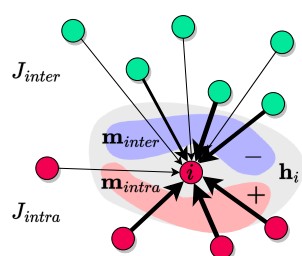

Figure 3: Illustration of the single-step message passing of iDist (lines 5-12 in Algorithm 1). A residue $i$ receives complementary distance-weighted messages $\mathbf{m}_{intra}$ and $\mathbf{m}_{inter}$ from all residues within the same protein $J_{intra}$ and other partners $J_{inter}$. The messages are aggregated into the embedding $\mathbf{h}_i$ and the procedure is repeated for each interface residue. iDist efficiently detects near-duplicate PPIs as the ones having similar averaged interface embeddings.

Lastly, in steps on lines 13-18, the interface representation $\mathbf{z}_\mathcal{I}$ is derived by averaging the hidden features across the individual chains and then across the interaction. As described in Algorithm 2, iDist then simply calculates the Euclidean distance between two representations $\mathbf{z}_\mathcal{I}$ and $\mathbf{z}_\mathcal{J}$ to compare two interfaces $\mathcal{I}$ and $\mathcal{J}$. The outlined procedure can be seen as an implicit approximation of the structural alignment as the resulting representations are $SE(3)$-invariant.

**iDist evaluation and threshold selection.** In order to evaluate the approximation performance of iDist, we compare it with the structural alignmnet algorithm iAlign (Gao & Skolnick, 2010a), the adaptation of TM-score (Zhang & Skolnick, 2004) to protein–protein interfaces. Figure 4 shows two main modes of the joint probability distribution of the scores output by both methods. For the scores where iAlign($\mathcal{I}, \mathcal{J}$) < 0.7 and iDist($\mathcal{I}, \mathcal{J}$) > 0.04, the interfaces $\mathcal{I}$ and $\mathcal{J}$ are typically distinct with rare cases of sharing similar structural patterns. In contrast, for the scores where iAlign($\mathcal{I}, \mathcal{J}$) > 0.7 and iDist($\mathcal{I}, \mathcal{J}$) < 0.04, the interfaces are very likely near duplicates. Notably, for this threshold of 0.04, iDist achieves 0.99% precision and 0.97% recall in detecting near-duplicate PPIs with iAlign score greater than 0.7. This observation suggests using the threshold of 0.04 for detecting near-duplicate PPIs with iDist.

To confirm the accurate detection of near-duplicate PPIs with iDist, we additionaly compare our method against independent USalign (Zhang et al., 2022) using the same benchmark employed to compare against iAlign. For this, we estimate the similarity score USalign($\mathcal{I}, \mathcal{J}$) by averaging two output TM-scores corresponding to querying PPI $\mathcal{I}$ against $\mathcal{J}$ and, vice versa, $\mathcal{J}$ against $\mathcal{I}$ with USalign. As a result of comparison, iDist under the same aforementioned threshold of 0.04 achieves 0.99% precision and 0.97% recall with respect to USalign under the threshold of 0.8 (Figure 6). This result agrees with the comparison of iDist against iAlign. Please note that the optimal near-duplicate thresholds of iAlign (0.7) and USalign (0.8) differ since the methods modify TM-score using different constants.

Based on the described evaluation, we use iDist with the threshold of 0.04 for detecting near-duplicate PPI interfaces that are defined by the distance of $6\mathring{A}$ between heavy atoms, in our analysis of DIPS composition. Nevertheless, we re-calibrate the threshold to 0.03 for the interfaces defined based on the $10\mathring{A}$ cutoff distance. This includes the construction of PPIRef50K, used for PPIFORMER pre-training.

**Construction of PPIRef300K.** Before applying iDist to deduplicate the protein–protein interactions mined from PDB (to construct PPIRef50K), we filter them to only preserve the proper interactions. Specifically, in order to extract only the biochemically proper protein–protein interactions from the raw PPIRef800K dataset and create PPIRef300K, we apply three standard filtering criteria (Townshend et al., 2019). An interaction passes the criteria if: (i) the structure determination

method is "x-ray diffraction" or "electron microscopy", (ii) the crystallographical resolution is 3.5Å or better, and (iii) the buried surface area (BSA) of the interface is at least 500Å². As a result, 99% of putative interactions satisfy the method criterion, 79% the resolution, and 47% the BSA. We calculate BSA using the dr_sasa software by Ribeiro et al. (2019).

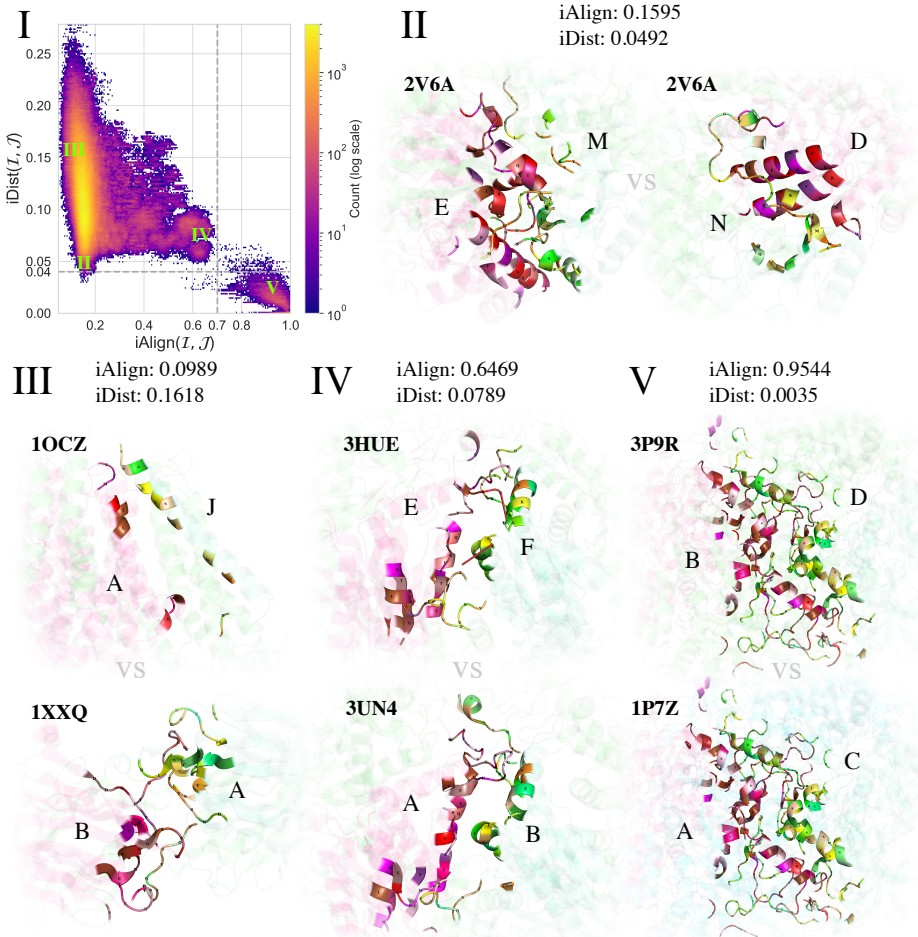

Figure 4: **Benchmarking our efficient iDist approximation of the structural alignment algorithm iAlign.** (I) Joint log-scale histogram displaying pair-wise iAlign (IS-score mode, 6Å cutoff) and iDist values of 1646 PPI interfaces (2,709,316 pairs) corresponding to 100 PDB codes sampled from DIPS (Townshend et al., 2019). The iAlign values vary between 0 and 1, with high values corresponding to well-aligned interfaces (1 for identical interfaces) and low values corresponding to poorly-aligned interfaces. The iDist varies between 0 to infinity with high values corresponding to structurally-distant interfaces and low values corresponding to similar interfaces (0 for identical interfaces). Figures (III, IV, V) depict samples from regions in (I) where the two methods agree, while (II) shows an example of a disagreement. Each figure displays two interfaces colored by amino acid types, with one protein's palette in reddish hues and the other one in greenish hues. (II) Example of disagreement. The score of iAlign corresponds to the expected value of the alignment of two random PPIs, while iDist suggests higher similarity due to the identity of several fragments of chains M and N (note the $\varepsilon$-like green loop and its further continuation) and similar compositions of amino acids in the helices belonging to proteins E and D (similar combination of reddish colors). In fact, the two interfaces represent different interaction modes of the same two chains in a large symmetric complex. (III) Unrelated interfaces. (IV) Interfaces on the edge of being considered near duplicates. The interactions are obviously related, but the geometry and primary structure differ at every local fragment. (V) Near duplicates. The proteins are visualized using PyMOL (DeLano et al., 2002).

## B    IMPLEMENTATION DETAILS

We implement PPIFORMER in PyTorch (Paszke et al., 2019) leveraging PyTorch Geometric (Fey & Lenssen, 2019), PyTorch Lightning (Falcon & The PyTorch Lightning team, 2019), Graphein (Jamasb et al., 2020), and a publicly available implementation of Equiformer[4]. We pre-train our model on four AMD MI250X GPUs (8 PyTorch devices) in a distributed data parallel (DDP) mode. Our best model was trained for 32 epochs of dynamic masking in 22 hours.

We pre-train PPIFORMER by sampling mini-batches of protein–protein interfaces using the Adam optimizer with the default $\beta_1 = 0.9, \beta_2 = 0.999$ (Kingma & Ba, 2014). We partially explore the grid of hyper-parameters given by Table 5, and select the best model according to the performance on zero-shot $\Delta\Delta G$ inference on the training set of SKEMPI v2.0. We further fine-tune the model on the same data with the learning rate of $3 \cdot 10^{-4}$ and sampling 32 mutations per GPU in a single training step, such that each mutation is from a different PPI. We employ the three-fold cross-validation setup discussed in the next section, and ensemble three corresponding fine-tuned models for test predictions. We observe that dividing $C_\alpha$ coordinates by 4, as proposed by Watson et al. (2023), increases the rate of convergence.

Table 5: Setting of the key pre-training hyper-parameters. The configuration of the best model, referred to as PPIFORMER, is highlighted in bold.

| Category | Hyper-parameter | Values |
|---|---|---|
| Data | Dataset | **PPIRef50K**, PPIRef300K, PPIRef800K, DIPS, iDist-deduplicated DIPS |
| | Num. neighbors $k$ | **10**, 30 |
| | Type-1 features $\mathbf{F}_1$ | $\{\mathbf{C}_\alpha \to \mathbf{C}_\beta^{\mathbf{virtual}}\}$, $\{C_\alpha \to C_\beta^{virtual}, C_\alpha \to N, C_\alpha \to C\}$, $\{C_\alpha \to C_\beta^{virtual}, C_\alpha \to C_\alpha^{prev}, C_\alpha \to C_\alpha^{next}\}$ |
| Masking | Fraction | $1, 15\%, 30\%, \mathbf{50\%}$ |
| | Same chain | **True**, False |
| | 80%10%10% BERT masking | **True**, False |
| Equiformer | Num. layers $L$ | 2, 4, **8** |
| | Num. heads $K$ | **2**, 4, 8 |
| | Hidden degree $deg$ | **1** |
| | Hidden dimensions $\mathbf{d}$ | $[\mathbf{128}, \mathbf{64}]$ |
| Loss | Label smoothing $\epsilon$ | $0, \mathbf{0.1}, 0.5$ |
| | Class weights $\mathbf{w}$ | **True**, False |
| Optimization | Learning rate | $1 \cdot 10^{-3}, \mathbf{5 \cdot 10^{-4}}, 3 \cdot 10^{-4}, 1 \cdot 10^{-4}$ |
| | Batch size per GPU | 8, 16, **32**, 64 |

## C    EXPERIMENTAL SETUP

This section provides additional details about the experiments with PPIFORMER. We describe the choice and setup of the datasets (Appendix C.1), metrics (Appendix C.2) and baseline models (Appendix C.3).

### C.1    TEST DATASETS

**SKEMPI v2.0.**    SKEMPI v2.0 (Jankauskaitė et al., 2019) is an invaluable resource for training and evaluating $\Delta\Delta G$ predictors. Since its initial release (Moal & Fernández-Recio, 2012), the dataset has played a pivotal role in advancing the field of computational PPI design (Geng et al., 2019b). However, despite encompassing mutations from nearly 300 available studies, it still only contains 7085 mutations across only 343 PPI structures. Moreover, the dataset is highly biased, with

---

[4]https://github.com/lucidrains/equiformer-pytorch

nearly three-quarters of the data corresponding to single-point mutations, more than half of which are mutations to alanine. Additionally, the dataset contains biases towards specific types of PPIs, including the repetition of near-duplicate structures or identical mutations with slightly different $\Delta\Delta G$ annotations. While it is difficult to enhance the quality or diversity of the dataset without additional wet lab experiments, it is crucial for the machine learning research community to leverage the dataset for training models that generalize beyond the available data. Here, we reconsider the data splitting strategy employed for training and evaluating state-of-the-art models and propose a new scheme based on an in-depth analysis of the original publication by Jankauskaitė et al. (2019).

In contrast to previous works (Luo et al., 2023; Shan et al., 2022; Liu et al., 2021), we do not use a PDB-disjoint split of SKEMPI v2.0 (based on the `#Pdb` column in the dataset) for training and assessing the performance of $\Delta\Delta G$ predictors. We find that this approach leads to at least two kinds of data leakage, potentially limiting the generalization of models towards unseen binders. To illustrate this leakage, we analyze a recent cross-validation split used to train and test the RDE-Network model (Luo et al., 2023).

First, we observe that every third test mutation in the split is, in fact, a mutation of the training PPI with a different PDB code in the test set (but with the same values in the `Protein 1` and `Protein 2` columns). For instance, PPIs with the `2WPT_A_B` and `1EMV_A_B` codes have nearly identical 3D structures, representing the same interaction between the colicin toxin and an immune protein of *E. coli*. These structures have 64 and 59 annotated mutations, respectively, with 21 mutations common to both structures. However, due to distinct `#Pdb` codes, `2WPT_A_B` and `1EMV_A_B` end up in different folds in (Luo et al., 2023), resulting in data leakage on the level of mutations.

Second, over half of the test `#Pdb` codes violate the original hold-out separation proposed by the authors of SKEMPI v2.0. Specifically, Jankauskaitė et al. (2019) performed automated clustering of PPIs followed by manual inspection to split entries into independent groups to assess machine learning generalization, resulting in the `Hold_out_proteins` column in the dataset. We observe that 57% of test PPIs in the PDB-disjoint split in (Luo et al., 2023) violate the proposed grouping, having a homologous PPI with the same `Hold_out_proteins` value in the training set. For example, the first two PPIs in the SKEMPI v2.0 table, `1CSE_E_I` and `1ACB_E_I`, are both interactions between a serine protease and the protein inhibitor Eglin c. These PPIs, coming from the same study (Qasim et al., 1997), have 6 identical annotated mutations of Eglin c with similar $\Delta\Delta G$ values. As a result, Jankauskaitė et al. (2019) assigned these PPIs to the same `Hold_out_proteins` group, `Pr/PI`, representing protease–protein inhibitor interactions. Nevertheless, since the PPIs have different PDB codes (`#Pdb`), they got separated into different folds in (Luo et al., 2023), which demonstrates the data leakage on the level of proteins.

Therefore, to ensure effective evaluation of generalization and mitigate the risk of overfitting, we divide SKEMPI v2.0 into 3 cross-validation folds based on the `Hold_out_proteins` feature, as originally proposed by the dataset authors. Additionally, we stratify the $\Delta\Delta G$ distribution across the folds to ensure balanced labels. Before constructing the cross-validation split, we reserve 5 distinct PPIs to create 5 test folds. We consider a PPI distinct if it does not share interacting partners or homologous binding site (Jankauskaitė et al., 2019) with any other PPI in SKEMPI v2.0[5] (and, consequently, has a unique `Hold_out_proteins` value). Furthermore, we ensure that for each of the 5 selected interactions, both negative and positive $\Delta\Delta G$ labels are present. Finally, the maximum iAlign IS-score between all PPIs in the test folds and those in the train-validation folds is 0.22, confirming the intended structural independence of the test set. Please refer to Table 8 for details on the constructed test set.

**Optimization of a human antibody against SARS-CoV-2.** We utilize the benchmark of Luo et al. (2023) to test the capabilities of models to optimize a human antibody against SARS-CoV-2 (Shan et al., 2022). Specifically, the goal is to retrieve 5 mutations on the heavy chain CDR region of the antibody that are known to enhance the neutralization effectiveness within a pool of all exhaustive 494 single-point mutations on the interface. Please note that the effects of the other 489 mutations are considered unknown and may be either favorable or unfavorable.

---

[5]Figure 2 in the original SKEMPI v2.0 paper (Jankauskaitė et al., 2019) illustrates such distinct PPIs by connected components consisting of two nodes (proteins), or more than two nodes but with two unique ones, where uniqueness is defined by the "Share common binding site" edges.

**Engineering staphylokinase for enhanced thrombolytic activity.** We assess the potential of the methods to enhance the thrombolytic activity of the staphylokinase protein (SAK). Staphylokinase, known for its cost-effectiveness and safety as a thrombolytic agent, faces a significant limitation in its widespread clinical application due to its low affinity to plasmin (Nikitin et al., 2022). In this study, we leverage 80 thrombolytic activity labels associated with SAK mutations located at the binding interface with plasmin (Laroche et al., 2000).

Specifically, we evaluate the $\Delta\Delta G$ predictions on the 1BUI structure from PDB, which contains the trimer consisting of plasmin-activated staphylokinase bound to another plasmin substrate. Unlike in the case of the SARS-CoV-2 benchmark, all 80 binary labels for SAK–plasmin have experimentally measured effects, among which 20 mutations are favorable. Additionally, 6 of them introduce at least two-fold thrombolytic activity improvement, constituting even more practically-significant targets for retrieval. Besides that, 24 out of 80 mutations on SAK are multi-point, which provides a more general setup for the evaluation than the SARS-CoV-2 benchmark. Note that while the quantity measured for SAK is the activity, what we are estimating is the $\Delta\Delta G$ for the SAK–plasmin interaction. Since the affinity of the complex is the main activity bottleneck, these two quantities were shown to be highly correlated (Nikitin et al., 2022).

## C.2 EVALUATION METRICS

In a practical binder design scenario (Nikitin et al., 2022; Shan et al., 2022), the primary objectives typically include (i) prioritizing mutations with lower $\Delta\Delta G$ values, and more specifically, (ii) identifying stabilizing mutations (with negative $\Delta\Delta G$) from a pool of candidates. To address (i), we evaluate models using the Spearman correlation coefficient between the ground-truth and predicted $\Delta\Delta G$ values. For (ii), we calculate precision and recall with respect to mutations with negative $\Delta\Delta G$. We calculate metrics for each protein–protein interaction separately, ensuring that mutations from different interactions are not combined. Subsequently, we average these results to obtain aggregated metric values. This approach estimates the expected performance of a model on a new, unseen PPI.

To enable comparisons with other methods, we also report metrics used in previous works: Pearson correlation coefficient, mean absolute error (MAE) and root mean squared error (RMSE), as well as area under the receiver operating characteristic (ROC AUC) with respect to the sign of mutations. However, we stress that these metrics may be misleading when selecting a model for a practical application. Pearson correlation may not capture the non-linear scoring capabilities of a method and is sensitive to outlying and less significant predictions on destabilizing mutations. MAE and RMSE are not invariant to monotonic transformations of predictions, and ROC AUC may overemphasize the performance on destabilizing mutations.

On the independent SARS-CoV-2 and SAK engineering case studies, we report scores for each of the favorable mutations following Luo et al. (2023). In addition, we evaluate more systematic precision at 1 (P@1) and precision at $k\%$ on the total pool of mutations (P@$k\%$) for $k \in \{5, 10\}$.

## C.3 BASELINES

**Flex ddG** (Barlow et al., 2018). Flex ddG is the most advanced Rosetta (Leman et al., 2020) protocol which predicts $\Delta\Delta G$ by estimating the change in binding free energies between the wild type and the mutant structures using force field simulations. The same protocol is used in a recent RosettaDDGPrediction toolbox (Sora et al., 2023). For each mutation, the $\Delta\Delta G$ prediction is obtained by averaging the output from 5 runs of `ddG-no_backrub_control`, a number shown to be optimal by Barlow et al. (2018). Since on average the prediction of a single $\Delta\Delta G$ on the SKEMPI v2.0 test folds using flex ddG requires approximately 1 CPU hour (on Intel Xeon Gold 6130), we do not evaluate the complete `ddG-backrub` protocol. When using the `ddG-backrub` parameters suggested by Barlow et al. (2018), the running time further increases by orders of magnitude, making the method impractical when compared to other baseline methods. For comparison, on the same data where flex ddG takes on average 1 CPU hour per mutation, our PPIFORMER requires 73 GPU milliseconds per mutation (using one out of two devices on AMD MI250X GPU with the batch size of 1).

**GEMME** (Laine et al., 2019). Global Epistatic Model for predicting Mutational Effects (GEMME) is a state-of-the-art sequence-based predictor that does not involve machine learning. The method explicitly models the evolutionary history of a sequence and derives scores of variants based on the evolutionary conservation of amino acids. GEMME is currently the second-best method in the ProteinGym benchmark[6] which was created to assess the performance of models in predicting protein fitness from deep mutations scanning data (Notin et al., 2022).

**MSA Transformer** (Rao et al., 2021). MSA Transformer is an unsupervised sequence-based baseline, trained in a self-supervised way on diverse multiple sequence alignments (MSA). We use the pre-trained model provided by Rao et al. (2021) and build MSAs against UniRef30 database using HHblits algorithm (Remmert et al., 2012) with the same parameters as were used to train the MSA Transformer. MSA Transformer can be applied to score mutations in a few-shot setup relying on masked pre-training (Meier et al., 2021). More specifically, the $\Delta\Delta G$ can be predicted as a difference of the log-likelihoods of the wild-type and mutant sequences conditioned on a provided MSA.

**ESM-IF** (Hsu et al., 2022). ESM-IF, an inverse folding model based in GVP-GNN (Jing et al., 2020), has been demonstrated by Hsu et al. (2022) to effectively generalize for predicting the signs of $\Delta\Delta G$ values for single-point mutations in SKEMPI (Jankauskaitė et al., 2019). To predict $\Delta G$, the authors compute the average log-likelihood of amino acids within a mutated chain, conditioned on the backbone structure of the complex. As done by the authors, to predict $\Delta\Delta G$, we subtract the likelihood of a mutant chain from that of the wild-type chain. Additionally, to account for simultaneous mutations across multiple chains, we perform an ESM-IF forward pass for each chain in a complex and average their likelihoods to obtain the final result. Our evaluation of ESM-IF on SKEMPI v2.0 leads to results consistent with the original publication. Specifically, ESM-IF achieves high ROC AUC of 0.68 on classifying the sign of mutations (Table 2), similarly to the ROC AUC of 0.71 reported for single-point mutations in (Hsu et al., 2022, Table C.5).

**RDE-Network** (Luo et al., 2023). To validate the performance of RDE-Network on our 5 test folds of SKEMPI v2.0, we exclude all test examples from cross-validation and retrain the model with the same hyperparameters as used by Luo et al. (2023). Since the exclusion of data points affects the sampling of training batches, and may, therefore, negatively affect the performance, we use 3 different random seeds and choose the final model according to the minimal validation loss. For the evaluation on SKEMPI-independent case studies, we use pre-trained model weights as provided by Luo et al. (2023).

## D  ABLATIONS

This section evaluates the effects of the key components proposed to enhance the generalization capabilities of PPIFORMER. We first demonstrate the importance of using the PPIRef50K dataset for pre-training, as well as employing the proposed pre-training regularization techniques (Appendix D.1). Next, in Appendix D.2 we demonstrate the positive impact of pre-training on the subsequent $\Delta\Delta G$ prediction task, and highlight the importance of the log odds ratio. Additionally, we verify the stability of the fine-tuning process across different random seeds.

### D.1  SELF-SUPERVISED PRE-TRAINING

Figure 5A (Top) illustrates that PPIFORMER achieves a notable performance without any supervised fine-tuning (per-PPI $\rho_{Spearman} = 0.21$ and precision = 35% on 5,643 mutations from SKEMPI v2.0). Next, we discuss the importance of the choice of the pre-training dataset used, as well as the proposed regularization measures.

**Pre-training dataset.** First of all, the best pre-training is achieved when sampling protein–protein interactions from our PPIRef50K dataset, both on scoring mutations and detecting the stabilizing ones (a3). We observe that training from redundant PPIRef300K decreases the performance, most probably because of introducing biases towards over-represented proteins and interactions into the

---

[6]https://proteingym.org/benchmarks

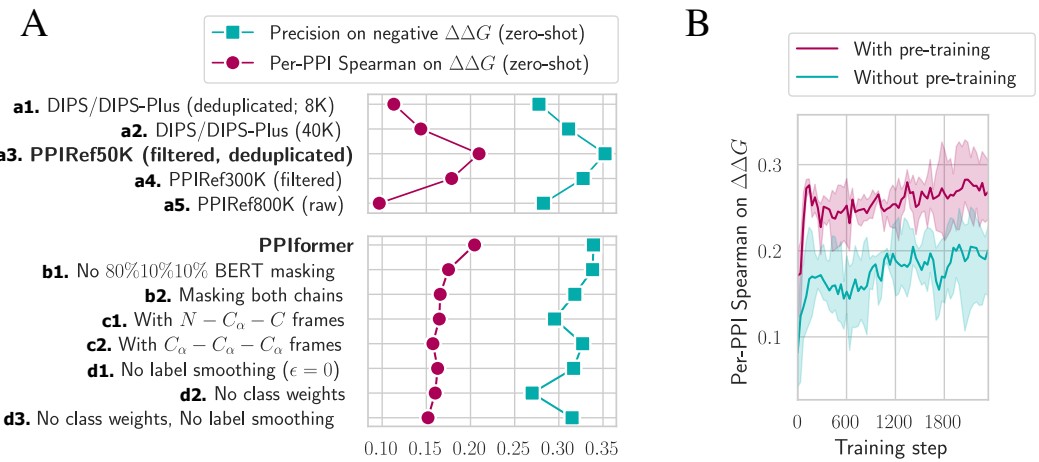

Figure 5: **Pre-training and fine-tuning ablations**. (A) Pre-training ablations with respect to datasets (a1-a5) and PPIFORMER masking strategies (b1-b2), input protein representations (c1-c2), and loss functions (d1-d3). The metrics show zero-shot performance on the training fold of the SKEMPI v2.0 dataset, i.e. $\Delta\Delta G$ inference according to Equation (8) without any supervision. Each row in the figure represents a single modification of the complete PPIFORMER model. Dataset ablations precede model ablations, using different SKEMPI v2.0 subsets. The dataset ablations (a1-a5) are done on two of the three folds from (Luo et al., 2023), and the model ablations (b1-b2) use the training set described in Appendix C.1. Thus, the performance varies slightly between (a3) and PPIFORMER rows (b1-b2), despite representing the same model. (B) The effect of pre-training on $\Delta\Delta G$ fine-tuning. Each training step is performed by sampling a single mutation from each PPI in a randomly sampled batch. The shaded areas correspond to the minimum, mean and maximum of the performance of three models in three-fold cross-validation.

model (a4). Finally, training from raw putative PPIs (a5) achieves the worst performance while training from DIPS or DIPS-Plus (a1, a2) is still worse than our PPIRef (a3).

**Pre-training regularization.** Next, we evaluate the benefits of three key components of PPIFORMER: masking strategy, data representation and the loss function. First, the $80\%10\%10\%$ masking (i.e. replacing 10% of masked nodes with native amino acids and 10% with random ones (Devlin et al., 2018)) leads to better performance (PPIFORMER vs. b1), as well as masking only residues from the same chain (PPIFORMER vs. b2), which better corresponds to practical binder design scenarios. Next, as discussed in Section 4.1, we represent the orientations of residues with a single virtual beta-carbon vector per residue. This is in contrast with the more widely adopted utilization of three vectors per residue describing the complete frame of an amino acid by including the direction to a real $C_\beta$ atom and the directions to alpha-carbons of neighboring residues (Jing et al., 2020) or, alternatively, two orthonormalized vectors in the directions of neighboring $N$ and $C$ atoms along the protein backbone (Watson et al., 2023; Yim et al., 2023). We observe that a single virtual beta carbon (used in our PPIFORMER) leads to better generalization compared to extra full-frame representations (c1, c2). Finally, the regularization by label smoothing and amino acid class weighting have additional strong positive impact on the generalization capabilities of PPIFORMER. Switching off the regularization measures significantly lowers the performance (d1-d3).

## D.2 FINE-TUNING FOR $\Delta\Delta G$ PREDICTION

**Effect of pre-training on fine-tuning.** We further ablate the importance of our self-supervised pre-training for the $\Delta\Delta G$ fine-tuning. Figure 5B illustrates the crucial significance of the pre-training. The pre-trained PPIFORMER surpasses the randomly initialized model with a margin of 0.08 absolute difference on per-PPI Spearman correlation. Note that the pre-trained zero-shot predictor (red curve at step 0) achieves performance competitive to the fully trained model without pre-training (green curve, highest peak).

**Fine-tuning head.**    In Section 4.4, we discuss the motivation of employing the log odds ratio for $\Delta\Delta G$ fine-tuning. The choice of log odds is strongly justified within the context of binding thermodynamics, offering the advantageous inductive bias of antisymmetry. Furthermore, this approach is compatible with the log-likelihood-based masking strategy employed during pre-training, ensuring a more effective transfer learning. We validate our argument by performing two ablation experiments.

First, following Liu et al. (2021), we implement a baseline regressor that performs two forward passes to predict $\Delta\Delta G$: one for the wild-type structure and another for a mutated structure (i.e., with replaced $\mathbf{F}_0$ features capturing one-hot encoded amino acid types). Then, it estimates $\Delta\Delta G$ by taking the average of node embeddings corresponding to residues being mutated in both structures, concatenating them, and applying a simple two-layer regressor of the form: $Linear(128, 64) \rightarrow ReLU \rightarrow Linear(64, 1)$. This approach results in a significant drop in performance (see the first row "Naive regressor" in Table 6).

Second, we improve this naive design by incorporating antisymmetry: removing biases in the $Linear$ layers, replacing $ReLU$ with $Tanh$ (an odd non-linear function), and subtracting averaged embeddings instead of concatenating them. This improvement leads to a performance boost compared to the naive baseline, but still strongly underperforms our log odds ratio (PPIFORMER) in terms of practically important metrics (see the second row "Antisymmetric regressor" in Table 6).

Table 6: Performance of PPIFORMER with different fine-tuning heads. The standard deviation for PPIFORMER is estimated from three fine-tuning experiments with different random seeds.

| Method | Spearman ↑ | Pearson ↑ | Precision ↑ | Recall ↑ | ROC AUC ↑ | MAE ↓ | RMSE ↓ |
|---|---|---|---|---|---|---|---|
| PPIFORMER (Naive regressor) | 0.298 | 0.330 | 0.204 | 0.258 | 0.724 | 1.844 | 2.210 |
| PPIFORMER (Antisymmetric regressor) | 0.348 | 0.382 | 0.494 | **0.626** | 0.658 | 1.720 | 1.964 |
| PPIFORMER (Ours, Section 4.4) | **0.44** ± 0.03 | **0.46** ± 0.03 | **0.60** ± 0.014 | 0.61 ± 0.012 | **0.78** ± 0.019 | **1.64** ± 0.011 | **1.94** ± 0.006 |

**Fine-tuning stability.**    Finally, we demonstrate that the test results are consistent under different random seeds used for fine-tuning (Table 7). Consequently, we report the average over the three different seeds and the standard deviation in other tables.

Table 7: Performance of PPIFORMER under different fine-tuning seeds.

| Method | Spearman ↑ | Pearson ↑ | Precision ↑ | Recall ↑ | ROC AUC ↑ | MAE ↓ | RMSE ↓ |
|---|---|---|---|---|---|---|---|
| PPIFORMER (Ours) | 0.44 ± 0.03 | 0.46 ± 0.03 | 0.60 ± 0.014 | 0.61 ± 0.012 | 0.78 ± 0.019 | 1.64 ± 0.011 | 1.94 ± 0.006 |
| PPIFORMER (Ours, seed 1) | 0.42 | 0.46 | 0.58 | 0.61 | 0.77 | 1.65 | 1.94 |
| PPIFORMER (Ours, seed 2) | 0.47 | 0.49 | 0.60 | 0.62 | 0.80 | 1.62 | 1.94 |
| PPIFORMER (Ours, seed 3) | 0.42 | 0.43 | 0.61 | 0.60 | 0.77 | 1.64 | 1.95 |

# E    ADDITIONAL RESULTS

**SKEMPI v2.0 test set.**    Table 2 shows the average performance of all compared methods on 5 test folds from SKEMPI v2.0. In this section, we provide a more detailed comparison. Specifically, Table 8 demonstrates non-aggregated performance of the methods on all test PPIs. Notably, PPIFORMER achieves state-of-the-art performance by attaining the best and second-best metric values in 7 and 4 cases, respectively, compared to the best competing methods GEMME (6 and 1) and RDE-Network (5 and 2).

Please note that in some cases methods fail to make $\Delta\Delta G$ predictions. For example, MSA Transformer fails to predict the effects of 4 mutations on the "dHP1 Chromodomain and H3 tail" interaction. This is caused by HHblits returing an empty MSA for the short binding peptide of 8 amino acids. To ensure a fair comparison, in such cases we impute predictions with the average $\Delta\Delta G$ value from the SKEMPI v2.0 training set, which is approximately 0.69.

| Method | Spearman | Precision | Recall |
|---|---|---|---|
| FLEX DDG | 0.68 | 100% | 50.00% |
| GEMME | 0.61 | **100%** | **100%** |
| MSA TRANSFORMER | 0.61 | **100%** | **100%** |
| ESM-IF | 0.34 | 50.00% | 50.00% |
| RDE-NET. | 0.68 | **100%** | **100%** |
| PPIFORMER | **0.75** | **100%** | **100%** |

Complement C3d and Fibrinogen-binding protein Efb-C (9 mutations, 4 neg., 5. pos.)

| Method | Spearman | Precision | Recall |
|---|---|---|---|
| FLEX DDG | 0.82 | 42.86% | 42.86% |
| GEMME | 0.40 | **100%** | 64.29% |
| MSA TRANSFORMER | 0.43 | 87.50% | 50.00% |
| ESM-IF | 0.18 | 41.18% | 50.00% |
| RDE-NET. | 0.58 | 42.11% | 57.14% |
| PPIFORMER | **0.60** | 38.46% | **71.43%** |

Barnase and barstar (105 mutations, 14 neg., 91 pos.)

| Method | Spearman | Precision | Recall |
|---|---|---|---|
| FLEX DDG | 0.98 | 100% | 100% |
| GEMME | **0.79** | **100%** | **80.00%** |
| MSA TRANSFORMER | 0.05 | 66.67% | 40.00% |
| ESM-IF | 0.09 | 0.00% | 0.00% |
| RDE-NET. | 0.15 | 50.00% | 40.00% |
| PPIFORMER | 0.34 | 60.00% | 60.00% |

C. thermophilum YTM1 and C. thermophilum ERB1 (10 mutations, 5 neg., 5 pos.)

| Method | Spearman | Precision | Recall |
|---|---|---|---|
| FLEX DDG | -0.05 | 29.27% | 85.71% |
| GEMME | -0.10 | 0.00% | 0.00% |
| MSA TRANSFORMER | 0.09 | 0.00% | 0.00% |
| ESM-IF | **0.10** | 45.45% | 71.43% |
| RDE-NET. | -0.40 | 30.23% | **92.86%** |
| PPIFORMER | 0.00 | **53.33%** | 57.14% |

dHP1 Chromodomain and H3 tail (46 mutations, 14 neg., 2 zero, 30 pos.)

| Method | Spearman | Precision | Recall |
|---|---|---|---|
| FLEX DDG | 0.29 | 44.44% | 33.33% |
| GEMME | 0.20 | 0.00% | 0.00% |
| MSA-TRANSFORMER | 0.37 | 0.00% | 0.00% |
| ESM-IF | 0.21 | 30.77% | **33.33%** |
| RDE-NET. | 0.21 | **50.00%** | **33.33%** |
| PPIFORMER | **0.43** | 40.00% | 16.67% |

E6AP and UBCH7 (49 mutations, 12 neg., 37 pos.)

Table 8: Performance of PPIFORMER and other competitive methods on 5 held-out test PPIs from the SKEMPI v2.0 dataset.

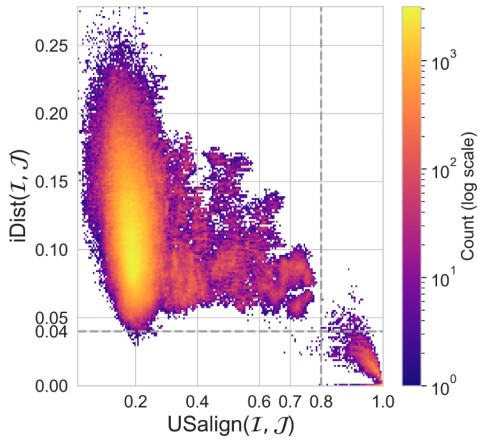

Figure 6: Extended panel for Figure 4 confirming that iDist accurately approximates not only iAlign but also independent USalign (Zhang et al., 2022) approach for detecting near-duplicate PPI structures. See Appendix A for details.

