# OpenReview forum: "Learning to design protein-protein interactions with enhanced generalization"
_ICLR.cc/2024/Conference — ICLR 2024 poster_

### Official Review · Reviewer_Sjjh · 2023-10-18

**Soundness:** 1 poor
**Presentation:** 3 good
**Contribution:** 2 fair
**Rating:** 3
**Confidence:** 5

**Summary:**

This work focuses on predicting binding affinity changes of protein complexes upon mutations, namely, it intends to solve $\Delta\Delta G$ prediction, which is critical in protein binder design. It has three major contributions: (1) They exhaustively mine PDB to build a large-scale non-redundant PPI dataset. (2) A novel transfer learning algorithm is introduced to bridge the conventional masked residue modeling and $\Delta\Delta G$ prediction. (3) A new state-of-the-art performance has been achieved at the standard Skempi2 dataset with a more reasonable data-splitting strategy. Overall speaking, I would recommend an acceptance to the ICLR committee but expect the author to elucidate more clearly about the new splitting mechanism.

**Strengths:**

(1) The paper built the largest and non-redundant dataset named PPIRef for representation learning of PPI.

(2) The study proposes a reasonable mechanism to transfer a masked language modeling (MLM)-pretrained PPI model to predict mutation effects. This abandons the traditional bidirectional forward scheme to enforce the antisymmetry.

(3) The author points out the flaws of previous methods' data splitting strategies and formulated a new one, which should be a good contribution to the machine learning community (but there are also some parts in the new splitting strategy that puzzle me, see questions).

(4) PPIFormer exhibits great potential in distinguishing the binding affinity change after mutations than existing algorithms such as RDE-Net and MSA-Transformer.

**Weaknesses:**

(1) My major concern is that neither the code nor the data is publicly available. Are authors willing to release them? If possible, I would recommend authors upload the necessary part of the method to an anonymous GitHub (https://anonymous.4open.science/) during the review phase.

(2) I value this study's effort in formulating a more comprehensive evaluation paradigm for mutation effect prediction, particularly in dataset splitting. However, the new splitting strategy is somehow unclear. For instance, the test set has only 5 samples, which are too small. And 3 cross-validation does not align with 5 test folds.

(3) In the task of retrieving 5 human antibody mutations effect against SARS-CoV-2, the performance of PPIFormer is not good enough, it only outperforms RDE-Net in 3 of 5 samples and is worse in P@5% and P@10% than RDE-Net.

**Questions:**

(1) I am a little confused about the evaluation of test folds. As mentioned in Appendix B.2, the author splits the dataset into 3 cross-validation folds based on the "hold-out-proteins feature" and then sets aside 5 random outlier PPIs to form 5 distinct test folds. Firstly, what is the "hold-out-proteins feature"? Please explain that term. Secondly, why are there 3 cross-validation folds but 5 test folds? Commonly, $k$ cross-validation would result in $k$ training sets as well as $k$ validation sets, and usually do not have a so-called test fold. In my understanding, these 5 extra outliers are excluded from both the training and validation sets and are used to evaluate models alone. So here comes the question: how does the author compute the $\Delta\Delta G$ for these 5 outliers? Notably, 3 cross-validation will lead to 3 sets of models that achieve the lowest losses in 3 different validation sets. Does the author just take an average of these 3 models' output to obtain the binding affinity change? If not, please specify the computational process.

Besides, how does the author split the Skempi2 by 3 cross-validation? I want to know more details. Also, why do these 5 outlier PPIs are selected randomly? I am afraid that a test set of merely 5 samples is so small. Is there any possibility to increase the test size?

(2) I understand that the splitting strategy of RDE-Net introduces data leakage. However, its 3 cross-validation is implemented without any technical mistakes. Therefore, can the author please just examine PPIFormer in the same splitting way as RDE-PPI and report the performance to help me better understand the superiority of PPIFormer over RDE-PPI?

Besides, can you report the performance of PPIFormer in the same way as RDE-Net? Namely, the Spearman and Pearson of PPIformer and different baselines in single-mutation and multiple-mutation cases. I hope to see that PPIFormer remains excellent in the multiple-mutation circumstance, which is more challenging and more likely to result in successful protein design.


(3) In the ablation study, the author claims the benefit of 80% 10% 10% masking proposed by BERT. However, I remember that BERT masked 15% of input tokens and tried to recover them. What does 80%, 10%, 10% mean?

What's more, in Figure (4), what is the difference between a3 and PPIFormer? It seems a3 is better than PPIFormer, but it is said PPIFormer is the best model. Please specify the difference between the upper and bottom subfigures.

(4) During the phase of mutant effect prediction, the author adopts a simple summation of all log odds ratios. In other words, each substitution of residue is considered independently. I suppose it may be a better solution to calculate the joint probability as $\log p (\hat{M} = M | \mathbf{c}_{\backslash M})$. Does the author agree with my point?

---

> ### Author Response · Authors · 2023-11-21
> **Official Comment by Authors (Part 1)**
>
> We thank the reviewer for the detailed analysis of our work and for providing us with highly valuable feedback.
>
> **Weaknesses**
> > (1) My major concern is that neither the code nor the data is publicly available. Are authors willing to release them? If possible, I would recommend authors upload the necessary part of the method to an anonymous GitHub (https://anonymous.4open.science/) during the review phase.
>
> Thank you for raising this point. We definitely plan to release all the code, trained models and data once they are  finalized and incorporate all comments / input from the review process.
>
> > (2) I value this study's effort in formulating a more comprehensive evaluation paradigm for mutation effect prediction, particularly in dataset splitting. However, the new splitting strategy is somehow unclear. For instance, the test set has only 5 samples, which are too small. And 3 cross-validation does not align with 5 test folds.
>
> We apologize for the confusion. Please see our response to question 1 for detailed clarification. The 5 test samples from SKEMPI v2.0 are difficult to expand due to the high redundancy of the dataset, and these samples are independent from the 3-fold cross-validation. We firstly set aside these 5 representative and independent PPIs for testing, and then split the remaining data into 3 folds.
>
> > (3) In the task of retrieving 5 human antibody mutations effect against SARS-CoV-2, the performance of PPIFormer is not good enough, it only outperforms RDE-Net in 3 of 5 samples and is worse in P@5% and P@10% than RDE-Net.
>
> Please note that while the SARS-CoV-2 benchmark proposed by [4] and [5] is an invaluable contribution to the field, one needs to be a bit careful making strong conclusions based on comparison of exact values on the benchmark. In this benchmark, only 5 of 494 mutations are annotated as favorable, while the effects of the other 489 are unknown. For example, according to the benchmark, RDE-Net retrieves the LH104F mutation with lower rank (the predicted rank is 5.47, which is better than the 10.93 predicted by PPIformer). However, we don’t know whether or not the hits predicted with better ranks by PPIformer are actually incorrect. Those mutations could also be favorable. The same holds true the other way around for high scoring mutations of RDE-Net. We will make a note about overinterpreting these results (both directions, in favour or against PPIFormer) in the paper.
>
> Additionally, there is a chance that the performance of RDE-Net, as well as DDGPred, on this benchmark may be slightly overoptimistic. The pool of the experimentally-validated mutations, including the 5 benchmarked ones, was preselected according to DDGPred predictions, and RDE-Net was trained in a similar way as DDGPred (i.e. on a PDB-disjoint split of SKEMPI v2.0). This means that the performance of the methods may be overoptimistic due to the potential circular data leakage. In other words, the methods are tested on the data, that were preselected by the same or a similar method.
>
> **Questions**
>
> > (1) I am a little confused about the evaluation of test folds. As mentioned in Appendix B.2, the author splits the dataset into 3 cross-validation folds based on the "hold-out-proteins feature" and then sets aside 5 random outlier PPIs to form 5 distinct test folds. Firstly, what is the "hold-out-proteins feature"? Please explain that term. Secondly, why are there 3 cross-validation folds but 5 test folds? Commonly,  cross-validation would result in  training sets as well as  validation sets, and usually do not have a so-called test fold. In my understanding, these 5 extra outliers are excluded from both the training and validation sets and are used to evaluate models alone. So here comes the question: how does the author compute the  for these 5 outliers? Notably, 3 cross-validation will lead to 3 sets of models that achieve the lowest losses in 3 different validation sets. Does the author just take an average of these 3 models' output to obtain the binding affinity change? If not, please specify the computational process
>
> We appreciate the detailed analysis of our data preparation and apologize for the confusion. We have now significantly expanded Appendix B.2 “Test Datasets” in the updated paper to address all the raised questions. The intuition of the reviewer about the parts that were unclear is correct, and we briefly answer the questions below.
>
> The “Hold out proteins” is [the column in the SKEMPI v2.0 dataset](https://life.bsc.es/pid/skempi2/database/browse), which specifies similar PPIs. We obtain our three folds by splitting the dataset by the values from this column. This ensures the PPIs are not leaked across folds. After 3-fold cross-validation, we, indeed, average three models for test predictions.

---

> > ### Author Response · Authors · 2023-11-21
> > **Official Comment by Authors (Part 2)**
> >
> > > Besides, how does the author split the Skempi2 by 3 cross-validation? I want to know more details. Also, why do these 5 outlier PPIs are selected randomly? I am afraid that a test set of merely 5 samples is so small. Is there any possibility to increase the test size?
> >
> > Prior to 3-fold splitting, we set aside 5 outlier PPIs for an independent test set. Please note that these outliers are not selected randomly. Instead, they are chosen from a limited pool of PPIs that do not have similar ones in the training data and that have both positive and negative ddG annotations.
> >
> > To elaborate, we manually examine the PPIs using the information in Figure 2 in the original SKEMPI v2.0 publication [6]. In this figure, connected components composed of two nodes (proteins) represent outlier PPIs. These PPIs do not share a similar binding site or partner with any other PPIs. Among these PPIs, there is a limited number that have both positive and negative ddG annotations. We were able to select 5 such PPIs only after inspecting the vast majority of outliers. This highlights the difficulty of choosing a larger, truly independent test set from SKEMPI v2.0 with a reasonable ddG distribution. We confirmed that the selected test PPIs do not have structural homologs among the other (training) PPIs by calculating interface similarities with iAlign. The maximum iAlign IS-score between all PPIs in the test folds and all PPIs in train-validation folds is 0.22.
> >
> > > (2) I understand that the splitting strategy of RDE-Net introduces data leakage. However, its 3 cross-validation is implemented without any technical mistakes. Therefore, can the author please just examine PPIFormer in the same splitting way as RDE-PPI and report the performance to help me better understand the superiority of PPIFormer over RDE-PPI?
> >
> > > Besides, can you report the performance of PPIFormer in the same way as RDE-Net? Namely, the Spearman and Pearson of PPIformer and different baselines in single-mutation and multiple-mutation cases. I hope to see that PPIFormer remains excellent in the multiple-mutation circumstance, which is more challenging and more likely to result in successful protein design.
> >
> > We have re-trained the same PPIformer architecture in the RDE-Net setup and examine its performance. We would like to emphasize that we use the same hyperparameters (e.g. architecture, loss and data representation) and do not tune the hyperparameters for these splits. We only adapt the batch size from 32 to 8 (we also explored different learning rates and batch sizes when experimenting with RDE-Net on out test set). Nevertheless, PPIformer yields a reasonable performance although it does not outperform supervised methods developed for PDB-disjoint splits. More specifically, PPIformer performs reasonably on the overall evaluation and on single-point mutations (although it is not the best method, but is the second best in some metrics). PPIformer improves on the challenging set-up of multi-point mutations where it is the best method according to some metrics, which may better reflect its generalization capabilities.   We want to emphasize that in this benchmark, the majority of test PPIs are leaked from the training set, and every third mutation is located on a PPI present in the training set (see Appendix B.2 “Test Datasets”).

---

> > > ### Author Response · Authors · 2023-11-21
> > > **Official Comment by Authors (Part 3)**
> > >
> > > Overall performance:
> > >
> > > |          |                | Per-Structure |           | Overall |           |        |        |        |
> > > |----------|----------------|---------|----------|---------|----------|--------|--------|--------|
> > > | **Category** | **Method**         | **Pearson** | **Spearman** | **Pearson** | **Spearman** | **RMSE**   | **MAE**    | **AUROC**  |
> > > | Sequence Based | ESM-1v        | 0.0073  | -0.0118  | 0.1921  | 0.1572   | 1.9609 | 1.3683 | 0.5414 |
> > > |              | PSSM           | 0.0826  | 0.0822   | 0.0159  | 0.0666   | 1.9978 | 1.3895 | 0.5260 |
> > > |              | MSA Transf.    | 0.1031  | 0.0868   | 0.1173  | 0.1313   | 1.9835 | 1.3816 | 0.5768 |
> > > |              | Tranception    | 0.1348  | 0.1236   | 0.1141  | 0.1402   | 2.0382 | 1.3883 | 0.5885 |
> > > | Energy Function | Rosetta       | 0.3284  | 0.2988   | 0.3113  | 0.3468   | 1.6173 | 1.1311 | 0.6562 |
> > > |                | FoldX         | 0.3789  | 0.3693   | 0.3120  | 0.4071   | 1.9080 | 1.3089 | 0.6582 |
> > > | Supervised   | DDGPred        | 0.3750  | 0.3407   | **0.6580** | 0.4687   | **1.4998** | **1.0821** | 0.6992 |
> > > |              | End-to-End     | 0.3873  | 0.3587   | 0.6373  | 0.4882   | 1.6198 | 1.1761 | 0.7172 |
> > > | Unsup./Semisup. | B-factor    | 0.2042  | 0.1686   | 0.2390  | 0.2625   | 2.0411 | 1.4402 | 0.6044 |
> > > |                | ESM-IF        | 0.2241  | 0.2019   | 0.3194  | 0.2806   | 1.8860 | 1.2857 | 0.5899 |
> > > |                | MIF-Dlogit    | 0.1585  | 0.1166   | 0.2918  | 0.2192   | 1.9092 | 1.3301 | 0.5749 |
> > > |                | MIF-Net.      | 0.3965  | 0.3509   | 0.6523  | 0.5134   | 1.5932 | 1.1469 | 0.7329 |
> > > | RDE-PPI      | RDE-Linear     | 0.2903  | 0.2632   | 0.4185  | 0.3514   | 1.7832 | 1.2159 | 0.6059 |
> > > |              | RDE-Net.       | **0.4448** | **0.4010** | 0.6447  | **0.5584** | $\underline{1.5799}$ | $\underline{1.1123}$ | **0.7454** |
> > > | Ours         | PPIformer      | $\underline{0.4281}$ | $\underline{0.3995}$ | $\underline{0.6450}$ | $\underline{0.5304}$ | 1.6420 | 1.1186 | $\underline{0.7380}$ |
> > >
> > > Performance on single-point mutations:
> > >
> > > |          |                | Per-Structure |           | Overall |           |        |        |        |
> > > |-----------------|------------------|---------|----------|---------|----------|--------|--------|--------|
> > > | **Category** | **Method**         | **Pearson** | **Spearman** | **Pearson** | **Spearman** | **RMSE**   | **MAE**    | **AUROC**  |
> > > | Sequence Based  | ESM-1v           | 0.0422  | 0.0273   | 0.1914  | 0.1572   | 1.7226 | 1.1917 | 0.5492 |
> > > |                 | PSSM             | 0.1215  | 0.1229   | 0.1224  | 0.0997   | 1.7420 | 1.2055 | 0.5659 |
> > > |                 | MSA Transf.      | 0.1415  | 0.1293   | 0.1755  | 0.1749   | 1.7294 | 1.1942 | 0.5917 |
> > > |                 | Tranception      | 0.1912  | 0.1816   | 0.1871  | 0.1987   | 1.7455 | 1.1708 | 0.6089 |
> > > | Energy Function | Rosetta          | 0.3284  | 0.2988   | 0.3113  | 0.3468   | 1.6173 | 1.1311 | 0.6562 |
> > > |                 | FoldX            | 0.3908  | 0.3640   | 0.3560  | 0.3511   | 1.5576 | 1.0713 | 0.6478 |
> > > | Supervised      | DDGPred          | 0.3711  | 0.3427   | 0.6515  | 0.4390   | 1.3285 | 0.9618 | 0.6858 |
> > > |                 | End-to-End       | 0.3818  | 0.3426   | $\underline{0.6605}$  | 0.4594   | $\underline{1.3148}$ | 0.9569 | 0.7019 |
> > > | Unsup./Semisup. | B-factor         | 0.1884  | 0.1661   | 0.1748  | 0.2054   | 1.7242 | 1.1889 | 0.6100 |
> > > |                 | ESM-IF           | 0.1616  | 0.1231   | 0.2548  | 0.1927   | 1.6928 | 1.1671 | 0.5630 |
> > > |                 | MIF-Dlogit       | 0.1053  | 0.0783   | 0.3358  | 0.2886   | 2.5361 | 1.8967 | 0.6066 |
> > > |                 | MIF-Net.         | 0.3952  | 0.3479   | **0.6667** | $\underline{0.4802}$   | **1.3052** | $\underline{0.9411}$ | 0.7175 |
> > > | RDE-PPI         | RDE-Linear       | 0.3192  | 0.2837   | 0.3796  | 0.3394   | 1.5997 | 1.0805 | 0.6027 |
> > > |                 | RDE-Net.         | **0.4687** | **0.4333** | 0.6421  | **0.5271** | 1.3333 | **0.9392** | **0.7367** |
> > > | Ours            | PPIformer        | $\underline{0.4192}$  | $\underline{0.3796}$ | 0.6287  | 0.4772   | 1.4232 | 0.9562 | $\underline{0.7213}$ |

---

> > > > ### Author Response · Authors · 2023-11-21
> > > > **Official Comment by Authors (Part 4)**
> > > >
> > > > Performance on multi-point mutations:
> > > > |                 |                  | Per-Structure |           | Overall |           |        |        |        |
> > > > |-----------------|------------------|---------|----------|---------|----------|--------|--------|--------|
> > > > | **Category**        | **Method**           | **Pearson** | **Spearman** | **Pearson** | **Spearman** | **RMSE**   | **MAE**    | **AUROC**  |
> > > > | Sequence Based  | ESM-1v           | -0.0599 | -0.1284  | 0.1923  | 0.1749   | 2.7586 | 2.1193 | 0.5415 |
> > > > |                 | PSSM             | -0.0174 | -0.0504  | -0.1126 | -0.0458  | 2.7937 | 2.1499 | 0.4442 |
> > > > |                 | MSA Transf.      | -0.0097 | -0.0400  | 0.0067  | 0.0030   | 2.8115 | 2.1591 | 0.4870 |
> > > > |                 | Tranception      | -0.0688 | -0.0120  | -0.0185 | -0.0184  | 2.9280 | 2.2359 | 0.4874 |
> > > > | Energy Function | Rosetta          | 0.1915  | 0.0836   | 0.1991  | 0.2303   | 2.6581 | 2.0246 | 0.6207 |
> > > > |                 | FoldX            | 0.2801  | 0.2771   | 0.2347  | 0.4137   | 2.5290 | 1.8639 | 0.6828 |
> > > > | Supervised      | DDGPred          | 0.3912  | 0.3896   | 0.5938  | 0.5150   | 2.1813 | 1.6699 | 0.7590 |
> > > > |                 | End-to-End       | $\underline{0.4178}$ | **0.4034** | 0.5858  | 0.4942   | 2.1971 | 1.7087 | 0.7532 |
> > > > | Unsup./Semisup. | B-factor         | 0.2078  | 0.1850   | 0.2009  | 0.2445   | 2.6557 | 2.0186 | 0.5876 |
> > > > |                 | ESM-IF           | 0.2016  | 0.1491   | 0.3260  | 0.3353   | 2.6446 | 1.9555 | 0.6373 |
> > > > |                 | MIF-Dlogit       | 0.1053  | 0.0783   | 0.3358  | 0.2886   | 2.5361 | 1.8967 | 0.6066 |
> > > > |                 | MIF-Net.         | 0.3968  | 0.3789   | 0.6139  | 0.5370   | $\underline{2.1399}$ | 1.6422 | 0.7735 |
> > > > | RDE-PPI         | RDE-Linear       | 0.1763  | 0.2056   | 0.4583  | 0.4247   | 2.4460 | 1.8128 | 0.6573 |
> > > > |                 | RDE-Net.         | **0.4233** | $\underline{0.3926}$ | $\underline{0.6288}$ | $\underline{0.5900}$ | **2.0980** | **1.5747** | $\underline{0.7749}$ |
> > > > | Ours            | PPIformer        | 0.3985  | 0.3925   | **0.6405** | **0.5946** | 2.1407 | $\underline{1.5753}$ | **0.7893** |
> > > >
> > > > > (3) In the ablation study, the author claims the benefit of 80% 10% 10% masking proposed by BERT. However, I remember that BERT masked 15% of input tokens and tried to recover them. What does 80%, 10%, 10% mean?
> > > >
> > > > BERT indeed masks 15% of tokens. However, for each of the masked tokens, it applies a random transformation as follows: with 80% probability, it leaves the masking token unchanged; with 10% probability, it replaces it with a random token; and with another 10% probability, it leaves the original token. This allows learning more powerful representations of non-masked tokens as it is more robust to noise [3].
> > > >
> > > > Following this idea, we adopt a similar approach. For 80% of the masked residues, we leave the F_0 features as zeros, and for the remaining 20%, we replace them with either the original amino acids or random amino acids. We hypothesize that in the case of proteins, this approach may not only lead to better embeddings of non-masked residues but also serve as a regularization simulating noisy random mutations.
> > > >
> > > > > What's more, in Figure (4), what is the difference between a3 and PPIFormer? It seems a3 is better than PPIFormer, but it is said PPIFormer is the best model. Please specify the difference between the upper and bottom subfigures.
> > > >
> > > > We apologize for the confusion. The ablations in the top figure (data ablations, including a3) were run prior to the ones in the bottom figure (model ablations, including PPIformer), using a slightly different subset of SKEMPI v2.0 for zero-shot evaluation. The data ablations were performed on training ddG examples corresponding to two out of three folds from the RDE-Net split [4], while the model ablations were run on the training set from our split. We updated the figure caption for clarification in the updated version of the paper.

---

> > > > > ### Author Response · Authors · 2023-11-21
> > > > > **Official Comment by Authors (Part 5)**
> > > > >
> > > > > > (4) During the phase of mutant effect prediction, the author adopts a simple summation of all log odds ratios. In other words, each substitution of residue is considered independently. I suppose it may be a better solution to calculate the joint probability as $\log{p(\hat{M} = M \vert \mathbf{c}_{\backslash M})}$. Does the author agree with my point?
> > > > >
> > > > > Yes, we agree that predicting the joint probability is highly beneficial over predicting the probability of each single-point substitution independently. The former can enable capturing epistasis (i.e. non-additiveness of mutation effects), which is an abundant phenomenon in proteins [1]. However, please note that in our model, each substitution is not considered independently during prediction. Instead, as shown in Figure 2, the probabilities are estimated in a single forward pass, and only then summed up for mutated positions. While this does not yield a probability from the exact joint conditional distribution $\log{p(\hat{{\bf c_{\it{M}}}} = {\bf c_{\it{M}}} \vert {\bf c_{\setminus \it{M}}})}$ (which is not obvious how to define for masked modeling [2]), one can show that during training the model learns to minimize the KL divergence between the distribution of predictions of masked residues $\sum_{i \in M}{\log{p(\hat{ c_{\it{i}}} = c_{\it{i}} \vert {\bf c_{\setminus \it{M}}})}}$ and the true joint conditional distribution $\log{q(\hat{{\bf c_{\it{M}}}} = {\bf c_{\it{M}}} \vert {\bf c_{\setminus \it{M}}})}$. Please see [2 (Appendix A)] for the derivation. This is an interesting point and we will discuss it in the revised paper.
> > > > >
> > > > > Please note that we slightly adapted the notation of the reviewer to match the rest of the formulas in our paper. We use $\log{p(\hat{{\bf c_{\it{M}}}} = {\bf c_{\it{M}}} \vert {\bf c_{\setminus \it{M}}})}$ instead of $\log{p(\hat{M} = M \vert \mathbf{c}_{\backslash M})}$, as the $M$ denotes a set of masked nodes, and $\mathbf{c}_M$ their values.
> > > > >
> > > > > **References**
> > > > >
> > > > > [1] Miton, Charlotte M., and Nobuhiko Tokuriki. "How mutational epistasis impairs predictability in protein evolution and design." Protein Science 25.7 (2016): 1260-1272.
> > > > >
> > > > > [2] Hennigen, Lucas Torroba, and Yoon Kim. "Deriving Language Models from Masked Language Models." arXiv preprint arXiv:2305.15501 (2023).
> > > > >
> > > > > [3] Devlin, Jacob, et al. "Bert: Pre-training of deep bidirectional transformers for language understanding." arXiv preprint arXiv:1810.04805 (2018).
> > > > >
> > > > > [4] Luo, Shitong, et al. "Rotamer Density Estimator is an Unsupervised Learner of the Effect of Mutations on Protein-Protein Interaction." bioRxiv (2023): 2023-02.
> > > > >
> > > > > [5] Shan, Sisi, et al. "Deep learning guided optimization of human antibody against SARS-CoV-2 variants with broad neutralization." Proceedings of the National Academy of Sciences 119.11 (2022): e2122954119.
> > > > >
> > > > > [6] Jankauskaitė, Justina, et al. "SKEMPI 2.0: an updated benchmark of changes in protein–protein binding energy, kinetics and thermodynamics upon mutation." Bioinformatics 35.3 (2019): 462-469.

---

> > > > > > ### Comment · Reviewer_Sjjh · 2023-11-22
> > > > > > **Reply by Sjjh**
> > > > > >
> > > > > > Thanks for your response and detailed explanation. I value your effort in trying to answering my questions. First of all, I want to emphasize my favor in your proposed PPIFormer, where a new iDist algorithm is invented to cluster interfaces and a transfer learning technique is introduced to align the pretrained model with ddG task. However, there are still several points that require further improvements and clarification.
> > > > > >
> > > > > > (1) I am glad that the author posted the performance of PPIFormer in the same split of RDE in Skempi. The results are not surprising. Because my team has reproduced nearly the same setting of yours (we created an interface dataset, used EquiFormer and adopted the same pretraining/fine-tuning techinque) but found it is impossible to outperform RDE. One of the key reason is that RDE is pretrained not solely on single-chains, but instead its crop of patches can also be located in interfaces (please see the code of RDE in "rde/utils/transforms/patch.py''). Moreover, we found EquiFormer fairly hard to search the hyper-parameters and cannot stack too many layers.
> > > > > >
> > > > > > Notably, the reported performance in RDE is not the unreachable. In the contrast, we discovered many simple mechanims to improve it and our new RDE can reaech a Spearman of 0.43 and a Pearson of 0.46. Though I know it is not reasonble to compare PPIFormer with ours, since we have not published or released any kind of paper work. But my major concern is that even though there are a lot new things in your study (new dataset, new iDist, new transfer learning, etc.), but the accumulated improvement is not significant. A recent paper DiffAffinity [A] evaluated their method on the same split of RDE and claimed better capability. I understand that DiffAffinity is only released recently and it is too harsh if we require you to compare PPIFormer with it. But I hold the view that if PPIFormer is adequatly strong, it should easily outperform RDE by a large margin even if the data split is not perfect.
> > > > > >
> > > > > > To summarize, I believe current results cannot fully convince me of the benefits of PPIFormer. There are many so-called innovations but its improvements are very limited over the baseline model RDE.
> > > > > >
> > > > > > [A] Predicting mutational effects on protein-protein binding via a side-chain diffusion probabilistic model. NIPS 2023
> > > > > >
> > > > > > (2) How do you fine-tune your Equiformer. To be specific, are you directly fine-tuning the entire model, or freeze the pretrained weight and regard it as a feature extractor like RDE?
> > > > > >
> > > > > > In addition, I have some suggestions for the authors. First, I think many of my questions do not need additional experiments, and just require clarificaiton. Since this year's discussion period is shorther than the last year, it is better to response as soon as possible rather than posting all answers at once. You replied at nearly the deadline of the discussion phase, and little time is left for our reviewers to exchange our point of views with you. Second, I do not agree with your idea that providing easily usable training code on such a short notice is challenging. Actually, we may not have enough time or computational resources to re-run your code (maybe some very interested reviewers will do). You do not need to upload the large interface data. What you have to do is to release the scripts of some of the key components of PPIFormer, so that we can directly go to your repo and see how you implemented it. Otherwise, we have to guess or ask in OpenReview how you designed and trained the model, which is not effecient. Experienced reviewers can immediately capture the innovations of your work via reviewing your code.
> > > > > >
> > > > > > Based on all these reasons, I would raise my score to 5 and wish PPIFormer good luck in the future.

---

> > > > > > > ### Author Response · Authors · 2023-11-22
> > > > > > > **Official Comment by Authors (Part 6)**
> > > > > > >
> > > > > > > We would like to thank the reviewer for their attention to our work. We appreciate their comments and respond to them below.
> > > > > > >
> > > > > > > We would like to emphasize that the goal of our work is to, first of all, develop a practically applicable method, i.e. a method that works on protein interactions unseen at training. With this goal, our evaluation covers 7 challenging protein-protein interactions from 3 different sources structurally distinct from the training data. Our results demonstrate state-of-the-art performance of PPIformer in such a setup simulating practical scenarios. We would like to respectfully point out that the strong claims of the reviewer regarding the performance of our method (e.g. "There are many so-called innovations but its improvements are very limited over the baseline model RDE.") ignore the key message of our work and are based solely on the highly leaking test data used in the RDE benchmark. In detail, we would like to again highlight that in the RDE benchmark [3], the majority of test PPIs are leaked from the training set, and every third mutation is located on a PPI present in the training set (please see Appendix B.2 “Test Datasets”). Below, we provide additional evidence that the RDE evaluation setup [3] (that is used by the reviewer as a measure of "performance", "improvment", or "results") may be misleading when developing methods for practical applications and, therefore, challenge the basis of the raised claims. Please note that we do not exclude that "the new RDE" developed and mentioned by the reviewer, which we understand is not yet published, may perform substantially better than the original version in our proposed non-leaking evaluation setup. Our main point is that to make progress on this problem the research community should consider non-leaking evaluation set-ups,  where the test protein interfaces are structurally different from the training set.
> > > > > > >
> > > > > > > > Notably, the reported performance in RDE is not the unreachable. In the contrast, we discovered many simple mechanims to improve it and our new RDE can reaech a Spearman of 0.43 and a Pearson of 0.46. Though I know it is not reasonble to compare PPIFormer with ours, since we have not published or released any kind of paper work. But my major concern is that even though there are a lot new things in your study (new dataset, new iDist, new transfer learning, etc.), but the accumulated improvement is not significant. A recent paper DiffAffinity [A] evaluated their method on the same split of RDE and claimed better capability. I understand that DiffAffinity is only released recently and it is too harsh if we require you to compare PPIFormer with it. But I hold the view that if PPIFormer is adequatly strong, it should easily outperform RDE by a large margin even if the data split is not perfect.
> > > > > > >
> > > > > > > We would appreciate if the reviewer had elaborated on the interpretation of the improvement in Spearman and Pearson metrics.  Can it be the case that these "simple mechanims" simply bias (non-robust) correlation coefficients to leaking outliers? Please consider, for example, the six most practically interesting predictions by RDE (i.e. the largest negative predictions of ddG; obtained with the official code from the RDE-PPI GitHub) :
> > > > > > >
> > > > > > > | Complex | Mutation | ddG | ddG prediction | Fold |
> > > > > > > |-|-|-|-|-|
> > > > > > > | 3BTD_E_I  | DI13K    | -11.5027 |  -10.3795  |      0 |
> > > > > > > | 3BTE_E_I  | EI13K    |  -9.4240 |   -9.3990 |      2 |
> > > > > > > | 3BTQ_E_I  | QI13K    | -9.3706 |   -8.8823 |      0 |
> > > > > > > | 3BTH_E_I  | HI13K    |  -8.7818 |   -8.8660 |      2 |
> > > > > > > | 3BTT_E_I  | TI13K    | -10.5975  |   -8.4427 |      1 |
> > > > > > > | 3BTW_E_I  | WI13K    |  -8.6641 |   -7.3462 |      2 |
> > > > > > >
> > > > > > > By examining the corresponding PPI structures, one can observe that all six examples represent six nearly identical protein-protein interactions (please see e.g. [3BTQ](https://www.rcsb.org/3d-view/3BTQ), [3BTD](https://www.rcsb.org/3d-view/3BTD)) and all labels characterize mutations on the same position and to the same amino acid type. Nevertheless, they are distributed across three separate train-validation-test folds according to the RDE benchmark. Such examples of leaks are abundant in SKEMPI v2.0, as discussed in Appendix B2 of our manuscript. Therefore, it is hard to conclude which of the methods is memorizing more training examples and which one is of a higher practical utility.

---

> > > > > > > > ### Author Response · Authors · 2023-11-22
> > > > > > > > **Official Comment by Authors (Part 7)**
> > > > > > > >
> > > > > > > > Additionally, we are strongly concerned about the fact that RDE-Linear, RDE-Net, as well as it seems DiffAffinity, are all evaluated on different 3-fold splits but compared together. By examining the almost identical data pre-processing source code for all three methods we found that (i) RDE-Linear was evaluated on 1405 less test samples but reported in the same test tables [3], (ii) data split is never fixed or stored on disk, instead it is determined by a random seed, (iii) random seed determining the split is set to different values for all three methods (compare for example the source codes of [RDE-Linear](https://github.com/luost26/RDE-PPI/blob/f6d68a7331a9f65290f525f0a491848245b072b3/rde/linear/calibrate.py#L477C5-L477C14), [RDE-Net](https://github.com/luost26/RDE-PPI/blob/f6d68a7331a9f65290f525f0a491848245b072b3/rde/datasets/skempi.py#L81C9-L81C25) and [DiffAffinity](https://github.com/EureKaZhu/DiffAffinity/blob/befbc610ca14982c9e7feea1485e01eb2252764e/context_generator/datasets/skempi.py#L84C6-L84C24)). We are afraid that such benchmarking may be largely a "competetition among random seeds for data splitting", especially considering the issues with leaking splitting discussed above. We strongly believe that our benchmarking approach provides more reliable way to compare models and therefore can unleash the full potential of our or other methods, including RDE and DiffAffinity. The issue of severe leakage between the training and test data is especially critical to understand and analyze considering the huge amount of effort and innovation put into these new, more advanced architectures worth of fair evaluation.
> > > > > > > >
> > > > > > > > > But my major concern is that even though there are a lot new things in your study (new dataset, new iDist, new transfer learning, etc.), but the accumulated improvement is not significant. A recent paper DiffAffinity [A] evaluated their method on the same split of RDE and claimed better capability. I understand that DiffAffinity is only released recently and it is too harsh if we require you to compare PPIFormer with it. But I hold the view that if PPIFormer is adequatly strong, it should easily outperform RDE by a large margin even if the data split is not perfect.
> > > > > > > >
> > > > > > > > Please note that our contributions primarily address the generalization to independent test data, which is orthogonal to optimizing machine learning architectures on given datasets. It is important to note that we are not aiming to outperform other methods on the leaking splits since we find that such evaluation does not fully reflect performance in practical applications (please also see Appendix B.2 for additional arguments and examples).
> > > > > > > >
> > > > > > > > > One of the key reason is that RDE is pretrained not solely on single-chains, but instead its crop of patches can also be located in interfaces (please see the code of RDE in "rde/utils/transforms/patch.py'').
> > > > > > > >
> > > > > > > > We would be grateful if the reviewer had provided some support for this claim. While it sounds like an important insight for our research domain, it is not obvious why training on PPIs+monomers rather than PPIs should be a "key" benefit for PPI design.
> > > > > > > >
> > > > > > > > > Moreover, we found EquiFormer fairly hard to search the hyper-parameters and cannot stack too many layers.
> > > > > > > >
> > > > > > > > Thank you for sharing with us your experience. Please note that the succesful applications of Equiformer use up to 20 layers [1, 2]. In our case, we also have not observed any such problems. Our Equiformer setup uses 8 layers (please see Table 5 in our submission for the list of all hyper-parameters in our setup).
> > > > > > > >
> > > > > > > > > (2) How do you fine-tune your Equiformer. To be specific, are you directly fine-tuning the entire model, or freeze the pretrained weight and regard it as a feature extractor like RDE?
> > > > > > > >
> > > > > > > > We fine-tune the entire model. It is justified by the close relationship between the masked modeling used in pre-training and the fine-tuning loss using the log odds ratio (please see Section 4.4). Please note that we do not add new parameters for fine-tuning.
> > > > > > > >
> > > > > > > > We highly appreciate the reviewer's suggestions regarding the organization of the code. We hope that our code that we have put on the Anonymised Github (as the reviewer suggested) addresses the raised concerns.
> > > > > > > >
> > > > > > > >  **References**
> > > > > > > >
> > > > > > > > [1] Liao, Yi-Lun, et al. "EquiformerV2: Improved Equivariant Transformer for Scaling to Higher-Degree Representations." arXiv preprint arXiv:2306.12059 (2023).
> > > > > > > >
> > > > > > > > [2] Lee, Jae Hyeon, et al. "Equifold: Protein structure prediction with a novel coarse-grained structure representation." bioRxiv (2022): 2022-10.
> > > > > > > >
> > > > > > > > [3] Luo, Shitong, et al. "Rotamer Density Estimator is an Unsupervised Learner of the Effect of Mutations on Protein-Protein Interaction." bioRxiv (2023): 2023-02.
> > > > > > > >
> > > > > > > > [4] Liu, Shiwei, et al. "Predicting mutational effects on protein-protein binding via a side-chain diffusion probabilistic model." arXiv preprint arXiv:2310.19849 (2023).

---

> > > > > > > > > ### Comment · Reviewer_Sjjh · 2023-11-23
> > > > > > > > >
> > > > > > > > > There are so many issues that I disagreed, but I do not want to talk much considering your attitude towards my suggestions.
> > > > > > > > >
> > > > > > > > > First, we strictly ran several different seeds to ensure our "simple techniques" is robustly beneficial instead of merely on one split. Please do not question our rigor in implementing experiments. Besides, who told you that RDE-Linear, RDE-Net, as well as it seems DiffAffinity, are all evaluated on different 3-fold splits but compared together? DiffAffinity used a different split strategy but it re-examined RDE as well. Therefore, you can see RDE only achieved a Spearman of 0.37 in its paper. Please read the paper carefully rather than simply draw the conclusion.
> > > > > > > > >
> > > > > > > > > Second, I still want to emphasize, I admit the existence of data leakage. But if your model is strong enough, you should outperform RDE-Net in whatever split. I have talked to RDE's authors before, and clearly know how they conducted the experiments. To be specific, RDE-Net used a relatively weak backbone (Graph Transformer), did not search the hyperparameters, and did not use complicated data like PPIRef. You brought up so many "new" things but still cannot beat it in standard split of RDE. But RDE performs so poorly in your "new" split with only 5 test samples. I have strong motivation to suspect that you reproduced RDE badly.
> > > > > > > > >
> > > > > > > > > Third, you mentioned a lot about the data leakage. But do you know that the Skempi v2 has many problems? Its value of ddG is not perfect? For instance, some complexes with the same mutations have different ddG (e.g., 3N01_A_B, 3NCB_A_B). Meanwhile,  some complex with different mutations have exactly the same ddGs (e.g., 2B2X_HL_A). There are, indeed, some errors or inaccuracy in the data. You can never train and test the data in a completely clean environment, especially for our AI for life science. A wonderful model should be competitive or the best in all circumstances. I really do not concur with the excuse that PPIFormer cannot outpass RDE due to the data leakage. You have already searched the hyperparameters, and you can use very small model to memorize the leakaged data, but why you still achieved low Spearman and Pearson?
> > > > > > > > >
> > > > > > > > > Fourth, I have already listed the clue for the authors to investigate the strategy of RDE in cropping patches ("rde/utils/transforms/patch.py'').  Can you please read the code first and ask for help? RDE randomly selects a chain and then found the closest 32 neighbors in this chain. After that, it choses the closest 96 residues to compose the final patch. These 96 residues can be located in any chains, and can be at the interface as well.
> > > > > > > > >
> > > > > > > > > | complex | mutstr | num_muts |  |
> > > > > > > > > | :--- | :--- | ---: | ---: |
> > > > > > > > > | 3NCB_A_B | AA167H | 1 | 0.686274 |
> > > > > > > > > | 3NCB_A_B | AA167H | 1 | -0.73409 |
> > > > > > > > > | 3NCB_A_B | AA167H | 1 | -1.66972 |
> > > > > > > > > | 3NCB_A_B | AA167H | 1 | -1.44501 |
> > > > > > > > > | 3NCB_A_B | AA167H | 1 | -1.48152 |
> > > > > > > > > | 3NCB_A_B | AA167H | 1 | -1.82152 |
> > > > > > > > > | 3NCB_A_B | AA167H | 1 | -1.63007 |
> > > > > > > > > | 3NCB_A_B | AA167H | 1 | -1.6149 |
> > > > > > > > > | 3NCB_A_B | AA167H | 1 | -1.53541 |
> > > > > > > > > | 3NCB_A_B | AA167H | 1 | -0.26778 |
> > > > > > > > > | 3NCB_A_B | AA167H | 1 | -0.77655 |
> > > > > > > > > | 3NCB_A_B | AA167H | 1 | -1.05533 |
> > > > > > > > > | 3NCB_A_B | AA167H | 1 | -1.71241 |
> > > > > > > > > | 3NCB_A_B | AA167H | 1 | -2.03769 |
> > > > > > > > > | 3NCB_A_B | AA167H | 1 | -2.20317 |
> > > > > > > > > | 3NCB_A_B | AA167H | 1 | -2.03356 |
> > > > > > > > > | 3NCB_A_B | AA167H | 1 | -2.45533 |
> > > > > > > > > | 3NCB_A_B | AA167H | 1 | -2.27295 |
> > > > > > > > > | 3NCB_A_B | AA167H | 1 | -2.21182 |
> > > > > > > > >
> > > > > > > > > | complex | mutstr | num_muts | ddG |
> > > > > > > > > | :--- | :--- | ---: | ---: |
> > > > > > > > > | 2B2X_HL_A | VH50T,EH64K,QL28S,YL52N | 4 | 1.043681 |
> > > > > > > > > | 2B2X_HL_A | VH50T,EH64K,QL28S,YL52N,FH99W | 5 | 1.043681 |
> > > > > > > > > | 2B2X_HL_A | VH50T,EH64K,QL28S,YL52N,GH54Y | 5 | 1.043681 |
> > > > > > > > > | 2B2X_HL_A | VH50T,EH64K,QL28S,YL52N,FH99W | 5 | 1.043681 |
> > > > > > > > > | 2B2X_HL_A | EH64K,QL28S,YL52N | 3 | 1.043681 |
> > > > > > > > > | 2B2X_HL_A | VH50T,EH64K,QL28S,YL52N,LH60D | 5 | 1.043681 |
> > > > > > > > > | 2B2X_HL_A | VH50T,QL28S,YL52N | 3 | 1.043681 |
> > > > > > > > > | 2B2X_HL_A | VH50T,EH64Q,QL28S,YL52N | 4 | 1.043681 |
> > > > > > > > > | 2B2X_HL_A | VH50T,EH64N,QL28S,YL52N | 4 | 1.043681 |
> > > > > > > > > | 2B2X_HL_A | VH50T,EH64K,QL28S | 3 | 1.043681 |
> > > > > > > > > | 2B2X_HL_A | VH50T,EH64K,QL28S,YL52E | 4 | 1.043681 |
> > > > > > > > > | 2B2X_HL_A | VH50T,EH64K,QL28S,YL52N,FH99W | 5 | 1.043681 |
> > > > > > > > > | 2B2X_HL_A | VH50T,EH64K,QL28S,YL52N,GH54Y | 5 | 1.043681 |
> > > > > > > > > | 2B2X_HL_A | VH50T,EH64K,QL28S,YL52N,FH99W | 5 | 1.043681 |
> > > > > > > > > | 2B2X_HL_A | QL28S,YL52N | 2 | 1.043681 |

---

### Official Review · Reviewer_Kuay · 2023-10-29

**Soundness:** 3 good
**Presentation:** 3 good
**Contribution:** 3 good
**Rating:** 6
**Confidence:** 2

**Summary:**

The manuscript presents a novel data splitting technique aimed at mitigating data leakage and generating a fresh dataset. Additionally, it introduces a new loss function for the equiformer, enhancing its ability to effectively undergo self-supervised training.

**Strengths:**

1. The paper is easy to read.

2. The authors present a novel splitting method to prevent data leakage in the PPI dataset.

3. They propose an innovative training scheme for EquiFormer, enabling self-supervised training.

4. Additionally, the paper introduces a new feature generation technique to capture more information both within and outside the residue itself.

5. Comprehensive experiments are conducted to evaluate the proposed method and the presented dataset.

**Weaknesses:**

When considering the data, it's important to note that the DIPS dataset may not have broad applicability for Protein-Protein Interaction (PPI) tasks. This is because PPI tasks involve not only binding, as observed in DIPS, but also encompass other aspects such as reaction, Ptmod, and activation labels, as described in reference [1].

[1] Learning Unknown from Correlations: Graph Neural Network for Inter-novel-protein Interaction Prediction

**Questions:**

1. In the optimization of a human antibody against SARS-CoV-2 and engineering staphylokinase for enhanced thrombolytic activity, how were the mutation pools obtained?

2. Can you explain the process used to identify the five PPI outliers mentioned in Datasets?

---

> ### Author Response · Authors · 2023-11-21
> **Official Comment by Authors**
>
> We appreciate the reviewer’s valuable feedback and address the comments below.
>
> **Weaknesses**
>
> > When considering the data, it's important to note that the DIPS dataset may not have broad applicability for Protein-Protein Interaction (PPI) tasks. This is because PPI tasks involve not only binding, as observed in DIPS, but also encompass other aspects such as reaction, Ptmod, and activation labels, as described in reference [1].
> [1] Learning Unknown from Correlations: Graph Neural Network for Inter-novel-protein Interaction Prediction
>
> We appreciate this important comment and agree that DIPS (as well as Protein Data Bank in general) does not extend beyond protein binding. Please, however, note that the focus of our paper is on designing protein-protein interactions for higher binding energy, and we do not study other, indirect ways of cooperation and mutual regulation between proteins. In this setup, the DIPS dataset (as well as our improved PPIRef) provides an important resource for self-supervised pre-training. Nevertheless, we believe that the reviewer’s comment is very stimulating for potential future work. Extending PPIformer to also learn from PPI networks, where nodes are proteins and edges are interaction types, may lead to improved performance, inspired by [1, 2]. We will cite and discuss the suggested reference.
>
> **Questions**
>
> > 1. In the optimization of a human antibody against SARS-CoV-2 and engineering staphylokinase for enhanced thrombolytic activity, how were the mutation pools obtained?
>
> For the SARS-CoV-2 test set, the authors of [3] selected a pool of all 494 possible single-point mutations on the heavy chain CDR region of the antibody. Out of these, five mutations were labeled as positive based on wet-lab experiments (please see [3] for the details on experimental setup). We obtained all the data from the official [RDE-PPI repository](https://github.com/luost26/RDE-PPI/tree/main).
>
> For the staphylokinase test set, we used the results of [4]. The authors of [4] measured the effects of mutations of staphylokinase on its thrombolytic activity. 80 of these annotated mutations are located at its interface with microplasmin, where 28 mutations reportedly increased the protein's thrombolytic activity, while the remaining 52 decreased it. We use these mutations as an another independent test set. Please see Appendix B.2 "Test datasets" in the updated paper for more details.
>
> > 2. Can you explain the process used to identify the five PPI outliers mentioned in Datasets?
>
> We select PPI outliers based on their similarity as defined by the authors of the SKEMPI v2.0 dataset [5]. Specifically, PPIs are considered similar if they share an interacting partner or a homologous binding site. Then, the outlier PPIs are defined as PPIs that do not have other similar PPIs. These outliers are depicted in Figure 2 of the original SKEMPI v2.0 publication [5] as connected components of two nodes (proteins), along with several other connected components mentioned in footnote 2 (on page 19) of our updated manuscript.
>
> We identified five outlier PPIs by manually analyzing these examples from the figure, aiming to select those with both positive and negative ddG annotations. We succeded at selecting the five outliers only after examining the vast majority of candidates, which highlights the difficulty of choosing a reasonable test set from highly biased SKEMPI v2.0. Lastly, we confirmed that the selected PPIs do not have structural homologs among the other (training) PPIs by calculating interface similarities with iAlign. For more details, please see Appendix B.2 "Test datasets" in the updated paper.
>
> **References**
>
> [2] Gao, Ziqi, et al. "Hierarchical graph learning for protein–protein interaction." Nature Communications 14.1 (2023): 1093.
>
> [3] Shan, Sisi, et al. "Deep learning guided optimization of human antibody against SARS-CoV-2 variants with broad neutralization." Proceedings of the National Academy of Sciences 119.11 (2022): e2122954119.
>
> [4] Laroche, Yves, et al. "Recombinant staphylokinase variants with reduced antigenicity due to elimination of B-lymphocyte epitopes." Blood, The Journal of the American Society of Hematology 96.4 (2000): 1425-1432.
>
> [5] Jankauskaitė, Justina, et al. "SKEMPI 2.0: an updated benchmark of changes in protein–protein binding energy, kinetics and thermodynamics upon mutation." Bioinformatics 35.3 (2019): 462-469.

---

### Official Review · Reviewer_brhp · 2023-10-30

**Soundness:** 3 good
**Presentation:** 3 good
**Contribution:** 2 fair
**Rating:** 6
**Confidence:** 4

**Summary:**

They first build an algorithm to approximate similarity between ppi interfaces and use it to assess the redundancy in existing datasets, like DIPS. Then they curate the available pdbs in protein data bank to create their dataset PPIREF, which reduces data leakage in train/test splits. The model architecture is comprised of Equiformer and it is first trained to predict masked residues with a regularised cross-entropy loss and then finetuned to predict the effects of mutations on binding affinity, using a thermodynamic-inspired loss.

**Strengths:**

- The paper is well-written and organized in a logical manner. It is easy to follow and the visualizations are informative.
- The authors put effort on choosing their metrics/loss functions/splits. Especially for the latter, they developed an algorithm to detect dataset redundancy and reduce data leakage in their splits. Overall, their design choices are well addressed.
- The model shows good performance on the benchmarks. it is nicely compared to different state of the art methods (both force field and ml methods) and using many metrics.

**Weaknesses:**

- IDIST could significantly improve the computation time needed to assess dataset redundancy, which can make redundancy tests a part of every dataset curation. However, I believe that establishing IDIST as part of dataset curation, requires more validation than comparing it against iAlign and for 100 pdbs. You should compare with more methods (eg Tm-align and US align) and repeat it for more pdbs (especially because you define the threshold of IDIST from the iAligh comparison)
- The novelty is a bit limited, especially in the model architecture.
- Appendix Figure 3/B: You mistakenly have the same pdb code in both subfigures.

**Questions:**

- I am worried that the F1 features introduce a data leakage and I would like to clarify it a bit:
Dauparas et al. describe utilizing distances between N, Cα, C, O, and a virtual Cβ placed based on the other backbone atoms, but their down-streaming task is predicting protein sequences in an autoregressive manner from the N to C terminus.
In your case, you mask a few amino acids, so with high probability neighbors of a masked residue will not be corrupted.
So I am worried that the F1 features can provide information regarding the "available space" in a certain position, biasing the model towards specific amino acids that fit well the available space. Is that a concern of yours?

---

> ### Author Response · Authors · 2023-11-21
> **Official Comment by Authors (Part 1)**
>
> We thank the reviewer for a detailed analysis of our work, and the valuable feedback! Below we address their comments.
>
> **Weaknesses**
>
> >IDIST could significantly improve the computation time needed to assess dataset redundancy, which can make redundancy tests a part of every dataset curation. However, I believe that establishing IDIST as part of dataset curation, requires more validation than comparing it against iAlign and for 100 pdbs. You should compare with more methods (eg Tm-align and US align) and repeat it for more pdbs (especially because you define the threshold of IDIST from the iAligh comparison)
>
> We appreciate the reviewer’s idea to establish iDist as a standard tool for dataset curation. We have performed a preliminary experiment according to the reviewer’s suggestion and demonstrate that iDist accurately approximates USalign as well (details below, and illustrated in the new supplementary **Figure 7** in the updated version of the manuscript). We plan to extend the benchmark to more than 100 PDB entries (2,7M PPI pairs) and potentially other methods for the camera-ready version of the paper. It was hard to do a large-scale analysis in the limited time of the rebuttal. Please note that we have not benchmarked iDist against TM-align, as suggested, because iAlign is an adaptation of TM-align to the domain of PPIs developed by the authors of TM-align (i.e. the group of Jeffrey Skolnick). Hence, we expect iAlign is better suited for PPIs than TM-align.
>
> In detail, iDist achieves high precision (99%) and recall (97%) with respect to near-duplicate PPI structures detected using USalign. Notably, this is a similar level of performance we achieved when benchmarking our iDist against iAlign. However, the iAlign and USalign methods modify TM-align differently, and with different constants, which introduces a shift in the marginal distributions of their alignment scores. We found that the threshold of 0.7 we used for the iAlign score corresponds approximately to a threshold of 0.8 on the USalign score (the threshold of iDist does not need to be adjusted for accurate approximation; see Figure 7 in the supplementary material). Regarding the computational time, we find that iDist is roughly 100 times faster than USalign, confirming the need for iDist for large-scale PPI analysis.
>
> >The novelty is a bit limited, especially in the model architecture.
>
> In our work, we have developed a new ddG predictor along with a reliable evaluation protocol. The novelty of our predictor (PPIformer) primarily lies in a newly developed data representation (coarse-grain representation based on virtual beta-carbons), training objectives (structural masked modeling for pre-training combined with log odds ratio for fine-tuning), and high-quality training data (PPIRef) prepared with our new method (iDist). To establish a new evaluation protocol, we have rethought the data splits and evaluation metrics that have been used for more than a decade (please see Section 2, Related Work for the references). Our evaluation scheme enables benchmarking ddG predictors in a way that goes beyond overfitting to near duplicate PPIs and is a step towards a new class of ddG predictors that will generalize to new unseen protein-protein interactions. We believe this work has the potential to significantly push forward the research community in this area.
>
> >Appendix Figure 3/B: You mistakenly have the same pdb code in both subfigures.
>
> Please note that this is not a mistake. Both PPI interfaces (E-M and N-D) come from the same PDB entry with the code [2V6A](https://www.rcsb.org/3d-view/2V6A). The complex is a 16-mer composed of two unique proteins. It is important to analyze the composition of interfaces within a single complex because symmetric complexes introduce a lot of redundancy (see also the caption under Figure 1 in the updated paper, which has been rewritten to be clearer).

---

> > ### Author Response · Authors · 2023-11-21
> > **Official Comment by Authors (Part 2)**
> >
> > **Questions**
> >
> > >I am worried that the F1 features introduce a data leakage and I would like to clarify it a bit: Dauparas et al. describe utilizing distances between N, Cα, C, O, and a virtual Cβ placed based on the other backbone atoms, but their down-streaming task is predicting protein sequences in an autoregressive manner from the N to C terminus. In your case, you mask a few amino acids, so with high probability neighbors of a masked residue will not be corrupted. So I am worried that the F1 features can provide information regarding the "available space" in a certain position, biasing the model towards specific amino acids that fit well the available space. Is that a concern of yours?
> >
> > We appreciate the reviewer's detailed analysis of our featurization. We indeed put high emphasis on designing F1 features such that they do not introduce data leakage.
> >
> > The studied training objective is to predict masked residues. In other words, our labels are the types of masked amino acids. The F1 features are independent of any specific amino acid types and solely capture the geometric properties of the backbones of interacting proteins. As the reviewer correctly describes, these features only consider the distances and angles between backbone atoms, being agnostic to side chains. Therefore, the F1 features do not introduce data leakage as they are agnostic to the labels (i.e. types of amino acids).
> >
> > Instead of introducing leakage, the F1 features provide a conditioning for predicting amino acids. Our rationale is that in protein interface engineering, where the goal is to enhance the affinity for a partner, it is typically desirable to increase binding energy while preserving other protein properties such as stability. Conditioning the model on a fixed backbone (and hence fold) can potentially lead to more successful design of protein binders. Therefore, the F1 features condition the predictions on the "available space" in terms of the backbone. However, it is important to note that the "available space" in terms of side chains is implicitly modeled to be flexible, in order to capture the diversity of possible conformations upon mutations. This is because we only capture the backbone geometry (coordinates X along with F1 features) and amino acid types (F0 features), rather than the precise atomic geometry of side chains or their chi angles. For example, consider a protein fragment consisting of three amino acids: glutamate-[MASK]-glutamate. While the prediction for the masked residue is constrained by the backbone through the F1 features, it is not constrained by the available space between side chains in the wild-type structure. Instead, the model can implicitly capture the flexibility of chi angles since it “does not see” the coordinates of glutamate side-chain atoms but only “knows” the types of the amino acids.

---

### Official Review · Reviewer_ZbAC · 2023-11-01

**Soundness:** 3 good
**Presentation:** 4 excellent
**Contribution:** 4 excellent
**Rating:** 8
**Confidence:** 5

**Summary:**

The authors construct PPIRef, the largest, non-redundant PPI dataset, using a proposed fast interface align method iDist for deduplication. The proposed SE(3)-equivariant model, PPIformer, based on Equiformer, is pretrained in a thermodynamics-motivated way on PPIRef and fine-tuned on SKEMPI. Experiments show that the pretrained PPIformer achieves new SOTA on non-leaking SKEMPI splits.

**Strengths:**

1. Novelty: A novel structural pretraining loss is proposed for mutation effect prediction.
2. Performance: New SOTA on SpearmanR on SKEMPI with non-leaking (harder) splits. Notably, this method does not rely on pre-computed mutant structures.
3. Clarity: The contributions are clearly stated and the paper is well-written.
4. Effort: The amount of work in this paper is quite comprehensive.

**Weaknesses:**

Please see questions.

**Questions:**

1. Would be nice if you could also compare your methods with baselines on the normal PDB-disjoint split, as this is also useful in real world cases. If PPIformer also performs best, the results would be more consistent and persuasive.
2. What is the logic behind your architecture choice? What is the size (#params) of your model?
3. In table 6, we can observe a high SpearmanR but the lower performance on precision and recall. What's your comment on this?
4. For real-world case studies, I suggest you try your model on deep mutational scanning (DMS) data, where the enrichment ratios of all single-point mutations are measured (though with higher noise). Ranking mutations on a single protein where only a few mutations have been experimentally measured is less convincing IMHO.

---

> ### Author Response · Authors · 2023-11-21
> **Official Comment by Authors (Part 1)**
>
> We greatly appreciate the valuable feedback provided by the reviewer. Below, we answer the reviewer's questions.
>
> **Questions**
> > 1. Would be nice if you could also compare your methods with baselines on the normal PDB-disjoint split, as this is also useful in real world cases. If PPIformer also performs best, the results would be more consistent and persuasive.
>
> We appreciate the reviewer's comment, but we would like to again emphasize that the PDB-disjoint splitting of SKEMPI v2.0 does not reflect the desired generalization in real-world scenarios. For instance, the first two PDB interaction codes present in the dataset, [1CSE_E_I](https://life.bsc.es/pid/skempi2/database/browse/mutations?keywords=mutations.pdb+contains+%221CSE_E_I%22+or+mutations.protein_1+contains+%221CSE_E_I%22+or+mutations.protein_2+contains+%221CSE_E_I%22&protein_search=1CSE_E_I) and [1ACB_E_I](https://life.bsc.es/pid/skempi2/database/browse/mutations?keywords=mutations.pdb+contains+%221ACB_E_I%22+or+mutations.protein_1+contains+%221ACB_E_I%22+or+mutations.protein_2+contains+%221ACB_E_I%22&protein_search=1ACB_E_I), come from different PDB entries ([1CSE](https://www.rcsb.org/structure/1CSE) and [1ACB](https://www.rcsb.org/structure/1ACB)) but have nearly identical 3D structures and sequences. Additionally, they are both labeled with the same six mutations, with slightly different ddG values. Despite this, they are divided into separate folds in the PDB-disjoint split by [5], exemplifying the data leakage problem. In SKEMPI v2.0, there are three times more unique PDB codes than PPIs, leading to most test PPIs having a near-duplicate PPI in the training set. Additionally, every third test mutation is located on a PPI present in the training set (please see Appendix B.2 "Test Datasets" for details).
>
> Following the request of the reviewer, we have now re-trained our identical PPIformer architecture in the RDE-Net setup and examine its performance below. We would like to emphasize that for our method we use the same hyperparameters (e.g. architecture, loss and data representation) and do not tune the hyperparameters for these leaked splits. We only adapt the batch size from 32 to 8 (we also experimented with different batch sizes when training RDE-Net on our splits). Overall, PPIformer has competitive performance. More specifically, PPIformer performs reasonably on the overall evaluation (although it is not the best method, it often is the second best). In addition, PPIformer is the best method according to some of the metrics on the challenging set-up of multi-point mutations, which may better reflect its generalization capabilities. Please also see the [review by Sjjh](https://openreview.net/forum?id=xcMmebCT7s&noteId=bfXV4RNeg5) highlighting the importance of multi-point mutations and our corresponding response for some other details.
>
> Overall performance:
> |          |                | Per-Structure |           | Overall |           |        |        |        |
> |----------|----------------|---------|----------|---------|----------|--------|--------|--------|
> | **Category** | **Method**         | **Pearson** | **Spearman** | **Pearson** | **Spearman** | **RMSE**   | **MAE**    | **AUROC**  |
> | Sequence Based | ESM-1v        | 0.0073  | -0.0118  | 0.1921  | 0.1572   | 1.9609 | 1.3683 | 0.5414 |
> |              | PSSM           | 0.0826  | 0.0822   | 0.0159  | 0.0666   | 1.9978 | 1.3895 | 0.5260 |
> |              | MSA Transf.    | 0.1031  | 0.0868   | 0.1173  | 0.1313   | 1.9835 | 1.3816 | 0.5768 |
> |              | Tranception    | 0.1348  | 0.1236   | 0.1141  | 0.1402   | 2.0382 | 1.3883 | 0.5885 |
> | Energy Function | Rosetta       | 0.3284  | 0.2988   | 0.3113  | 0.3468   | 1.6173 | 1.1311 | 0.6562 |
> |                | FoldX         | 0.3789  | 0.3693   | 0.3120  | 0.4071   | 1.9080 | 1.3089 | 0.6582 |
> | Supervised   | DDGPred        | 0.3750  | 0.3407   | **0.6580** | 0.4687   | **1.4998** | **1.0821** | 0.6992 |
> |              | End-to-End     | 0.3873  | 0.3587   | 0.6373  | 0.4882   | 1.6198 | 1.1761 | 0.7172 |
> | Unsup./Semisup. | B-factor    | 0.2042  | 0.1686   | 0.2390  | 0.2625   | 2.0411 | 1.4402 | 0.6044 |
> |                | ESM-IF        | 0.2241  | 0.2019   | 0.3194  | 0.2806   | 1.8860 | 1.2857 | 0.5899 |
> |                | MIF-Dlogit    | 0.1585  | 0.1166   | 0.2918  | 0.2192   | 1.9092 | 1.3301 | 0.5749 |
> |                | MIF-Net.      | 0.3965  | 0.3509   | 0.6523  | 0.5134   | 1.5932 | 1.1469 | 0.7329 |
> | RDE-PPI      | RDE-Linear     | 0.2903  | 0.2632   | 0.4185  | 0.3514   | 1.7832 | 1.2159 | 0.6059 |
> |              | RDE-Net.       | **0.4448** | **0.4010** | 0.6447  | **0.5584** | $\underline{1.5799}$ | $\underline{1.1123}$ | **0.7454** |
> | Ours         | PPIformer      | $\underline{0.4281}$ | $\underline{0.3995}$ | $\underline{0.6450}$ | $\underline{0.5304}$ | 1.6420 | 1.1186 | $\underline{0.7380}$ |

---

> > ### Author Response · Authors · 2023-11-21
> > **Official Comment by Authors (Part 2)**
> >
> > Performance on multi-point mutations:
> >
> > |                 |                  | Per-Structure |           | Overall |           |        |        |        |
> > |-----------------|------------------|---------|----------|---------|----------|--------|--------|--------|
> > | **Category**        | **Method**           | **Pearson** | **Spearman** | **Pearson** | **Spearman** | **RMSE**   | **MAE**    | **AUROC**  |
> > | Sequence Based  | ESM-1v           | -0.0599 | -0.1284  | 0.1923  | 0.1749   | 2.7586 | 2.1193 | 0.5415 |
> > |                 | PSSM             | -0.0174 | -0.0504  | -0.1126 | -0.0458  | 2.7937 | 2.1499 | 0.4442 |
> > |                 | MSA Transf.      | -0.0097 | -0.0400  | 0.0067  | 0.0030   | 2.8115 | 2.1591 | 0.4870 |
> > |                 | Tranception      | -0.0688 | -0.0120  | -0.0185 | -0.0184  | 2.9280 | 2.2359 | 0.4874 |
> > | Energy Function | Rosetta          | 0.1915  | 0.0836   | 0.1991  | 0.2303   | 2.6581 | 2.0246 | 0.6207 |
> > |                 | FoldX            | 0.2801  | 0.2771   | 0.2347  | 0.4137   | 2.5290 | 1.8639 | 0.6828 |
> > | Supervised      | DDGPred          | 0.3912  | 0.3896   | 0.5938  | 0.5150   | 2.1813 | 1.6699 | 0.7590 |
> > |                 | End-to-End       | $\underline{0.4178}$ | **0.4034** | 0.5858  | 0.4942   | 2.1971 | 1.7087 | 0.7532 |
> > | Unsup./Semisup. | B-factor         | 0.2078  | 0.1850   | 0.2009  | 0.2445   | 2.6557 | 2.0186 | 0.5876 |
> > |                 | ESM-IF           | 0.2016  | 0.1491   | 0.3260  | 0.3353   | 2.6446 | 1.9555 | 0.6373 |
> > |                 | MIF-Dlogit       | 0.1053  | 0.0783   | 0.3358  | 0.2886   | 2.5361 | 1.8967 | 0.6066 |
> > |                 | MIF-Net.         | 0.3968  | 0.3789   | 0.6139  | 0.5370   | $\underline{2.1399}$ | 1.6422 | 0.7735 |
> > | RDE-PPI         | RDE-Linear       | 0.1763  | 0.2056   | 0.4583  | 0.4247   | 2.4460 | 1.8128 | 0.6573 |
> > |                 | RDE-Net.         | **0.4233** | $\underline{0.3926}$ | $\underline{0.6288}$ | $\underline{0.5900}$ | **2.0980** | **1.5747** | $\underline{0.7749}$ |
> > | Ours            | PPIformer        | 0.3985  | 0.3925   | **0.6405** | **0.5946** | 2.1407 | $\underline{1.5753}$ | **0.7893** |
> >
> > > 2. What is the logic behind your architecture choice? What is the size (#params) of your model?
> >
> > The key logic behind our architecture is to maximize generalization of the model to unseen diversity of protein-protein interactions (PPIs). Our goal is to capture interaction patterns in potentially flexible and mutated protein-protein interfaces. To achieve this, we have chosen an architecture capable of (i) capturing coarse-grain representations of PPIs to implicitly model side-chain flexibility, (ii) effective pre-training from large unlabeled data, and (iii) seamless fine-tuning on small labeled data. The encoder f, represented by Equiformer, effectively achieves (i). The combination of the classifier g and masked modeling enables (ii). Finally, the log odds ratio enables (iii) without introducing any additional parameters (see Equation 8).
> >
> > Our model has 12,333,366 parameters.
> >
> > > 3. In table 6, we can observe a high SpearmanR but the lower performance on precision and recall. What's your comment on this?
> >
> > We can observe that, while PPIformer does not achieve the highest precision or recall in several cases, it consistently achieves reliable performance across all five test PPIs. This contrasts with other methods that strongly fail on one of the PPIs. This is also reflected in PPIformer having the highest values on precision and AUROC in the aggregated Table 2, and the second best value on the less important recall (Precision is prioritized over recall because, while it is typically relatively easy to find a favorable mutation for a protein, we need to be confident in the discovery).

---

> > > ### Author Response · Authors · 2023-11-21
> > > **Official Comment by Authors (Part 3)**
> > >
> > > > 4. For real-world case studies, I suggest you try your model on deep mutational scanning (DMS) data, where the enrichment ratios of all single-point mutations are measured (though with higher noise). Ranking mutations on a single protein where only a few mutations have been experimentally measured is less convincing IMHO.
> > >
> > > We agree with this valuable comment. While preparing the paper, we explored this direction but were unable to find larger and high-quality datasets of labeled protein-protein interaction variants. For instance, the ProteinGym benchmark includes 87 DMS assays, where the experimentally measured fitness is complex and, therefore, treated as a black box [2]. This means that without a detailed manual analysis of each assay, it is difficult to identify assays that are directly representative of  protein-protein interactions. Furthermore, even if relevant assays are found, it is unclear whether the corresponding interactions have solved 3D structures, hampering the application of structure-based methods (such as ours). Nevertheless, while we cannot compare ddG predictors on DMS data, we compare the best available predictor trained on DMS from ProteinGym data on the task of ddG prediction. We find that it performs worse than our method (please see our comment on the Weakness 2 by the [hYdu reviewer](https://openreview.net/forum?id=xcMmebCT7s&noteId=0tshmegDww) for the details).
> > >
> > > Additionally, previous work has already attempted to evaluate ddG predictors using deep sequencing data. Specifically, the DGCddG model was evaluated on the T56 target from the CAPRI ROUND 26 benchmark from 2012 [3]. The benchmarked complex includes 855 annotated mutations, and DGCddG achieved a near-zero correlation. The poor performance could be due to either the low quality of the data or poor generalization of the model (which is not state-of-the-art and is overfitted to a random split of mutations). Considering the near-zero correlation of all other evaluated methods strongly suggests the validity of the first point. We hypothesize that the measured fitness for T56 may not necessarily depend on the affinity of the studied complex, and the computationally constructed structure may have low quality.
> > >
> > > In conclusion, we find that currently available deep mutation scanning data is not suitable for training and evaluating the effects of mutations on protein-protein interactions. However, we believe that curating such datasets is a highly promising direction for the next generation of ddG predictors, especially considering recent advances in high-throughput assays [4]. If the reviewer is aware of any relevant work that we may have missed, we would be very grateful for providing us with the reference. We will incorporate it in the paper.
> > >
> > > **References**
> > >
> > > [1] Devlin, Jacob, et al. "Bert: Pre-training of deep bidirectional transformers for language understanding." arXiv preprint arXiv:1810.04805 (2018).
> > >
> > > [2] Notin, Pascal, et al. "Tranception: protein fitness prediction with autoregressive transformers and inference-time retrieval." International Conference on Machine Learning. PMLR, 2022.
> > >
> > > [3] Jiang, Yelu, et al. "DGCddG: Deep Graph Convolution for Predicting Protein-Protein Binding Affinity Changes Upon Mutations." IEEE/ACM Transactions on Computational Biology and Bioinformatics (2023).
> > >
> > > [4] Baryshev, Alexander, et al. "Massively parallel protein-protein interaction measurement by sequencing (MP3-seq) enables rapid screening of protein heterodimers." bioRxiv (2023): 2023-02.
> > >
> > > [5] Luo, Shitong, et al. "Rotamer Density Estimator is an Unsupervised Learner of the Effect of Mutations on Protein-Protein Interaction." bioRxiv (2023): 2023-02.

---

> > > > ### Comment · Reviewer_ZbAC · 2023-11-22
> > > > **Response to the Authors by Reviewer ZbAC**
> > > >
> > > > Thank you for your detailed and insightful responses, especially those for questions 1 and 4.
> > > > After reading your rebuttal and other reviewer's opinions, I have decided to maintain my score of 8. Though many points raised by reviewer Sjjh and others are valid and important (I also strongly suggest you open-source the code), I believe that the novelty and effort in this paper makes it qualified for an ICLR publication.

---

### Official Review · Reviewer_hYdu · 2023-11-01

**Soundness:** 3 good
**Presentation:** 3 good
**Contribution:** 3 good
**Rating:** 6
**Confidence:** 3

**Summary:**

This article established a machine learning model to predict protein-protein interactions. The authors constructed a three-dimensional protein-protein interaction database and pre-trained the model based on it. They then made predictions on two existing examples and compared the performance with other baseline methods, yielding better performance.

**Strengths:**

Protein-protein interaction is important for protein design and engineering. The authors established a new machine learning model to predict the $\Delta\Delta G$. The paper is well-organized and easy to follow.

**Weaknesses:**

1. Evaluations on a broader range of datasets are preferred in terms of PPI and mutational effect predictions. Currently, all the test samples are selected from a larger benchmark dataset, making it skeptical that the results might be cherry-picked.
2. Comparison of baseline methods is limited. At least some other deep learning methods should be included. For instance, for mutational effect prediction, the authors might refer to https://github.com/OATML-Markslab/ProteinGym.
3. The prediction results are not reported with standard deviation, which makes it hard to tell whether the performance is statistically significant.

**Questions:**

1. Would it be possible to construct the message passing between interfaces with GVP, another MPNN model that aggregates with protein scalar and vector features?
2. Why use log odds ratio for predicting $\Delta\Delta G$, instead of training a regressor to directly predicting the $\Delta\Delta G$ values?

---

> ### Author Response · Authors · 2023-11-21
> **Official Comment by Authors (Part 1)**
>
> We thank the reviewer for their constructive and valuable feedback on our work. Below, we address the reviewer’s comments.
>
> **Weaknesses**
>
> > 1. Evaluations on a broader range of datasets are preferred in terms of PPI and mutational effect predictions. Currently, all the test samples are selected from a larger benchmark dataset, making it skeptical that the results might be cherry-picked.
>
> Thank you for this important comment. Please note that in our work we use three independent test datasets (see Section 5.1 Datasets). First, we select 5 outlier (i.e. not having a training homologue) test PPIs from the SKEMPI v2.0 dataset. SKEMPI v2.0 encompasses the vast majority of published experimental ddG studies. This dataset, indeed, represents a selection from a larger benchmark dataset. We selected the test samples once to evaluate the generalization of our model before seeing the results. We did not overfit our method to the test set by exploring any other subsets. Please see Appendix B.2 “Test Datasets”, which now describes in detail the construction of the SKEMPI v2.0 test set.
>
> Moreover, we evaluate our method on two additional (and separate) datasets, which are not part of SKEMPI v2.0 and come from independent studies. The “Optimization of a human antibody against SARS-CoV-2” provides an independent evaluation on the benchmark proposed by [1] and [2], the studies proposing DDGPred and RDE-Network (ICLR 2023). Finally, to compare generalization of different methods even more robustly, we curated a novel independent test set for the problem of  “Engineering staphylokinase for enhanced thrombolytic activity”. This dataset was mined from [12].
>
> We hope that our test setup involving three mutually independent test sets provides enough evidence that the results are not cherry-picked. Please also see our comment on attempts to obtain more test data in the answer to the [ZbAC reviewer’s Question 4](https://openreview.net/forum?id=xcMmebCT7s&noteId=jfNTtLMy5P). Overall, it is challenging to find additional independent and high-quality test data.
>
> > 2. Comparison of baseline methods is limited. At least some other deep learning methods should be included. For instance, for mutational effect prediction, the authors might refer to https://github.com/OATML-Markslab/ProteinGym.
>
> Thank you for your important comment. Please note that, while the number of evaluated baselines is not excessive, we have initially selected methods among the most representative methods in four available categories:
> 1. Physics-based simulators. Flex ddG is the most performant and elaborate force-field-based protocol for binding ddG prediction.
> 2. Evolutionary methods. MSA-Transformer is a well-established method that might benefit from using multiple sequence alignment (MSA) compared to other baselines (MSA is crucial, for example, for AlphaFold2).
> 3. Unsupervised models. ESM-IF is a state-of-the-art inverse folding method, trained on the extensive AlphaFold DB. It was shown to effectively score single-point mutations from SKEMPI v2.0.
> 4. Supervised models. RDE-Network is a recent, high-quality, and state-of-the-art method fine-tuned on SKEMPI v2.0.
>
> We value the suggestion to consider additional baselines from ProteinGym. Consequently, we have now added the evaluation of GEMME, the best publicly available model in [ProteinGym](https://www.proteingym.org/substitutions) (GEMME ranks second among 34 methods and is publicly available, unlike the top-ranked model, TranceptEVE). We find that GEMME performs worse on most metrics than our PPIformer (please refer to the table below). However, it is worth noting that GEMME, as well as other ProteinGym methods, was not trained for PPI design. Instead, it was trained to optimize the global fitness of a protein with respect to deep mutational scanning experiments.

---

> > ### Author Response · Authors · 2023-11-21
> > **Official Comment by Authors (Part 2)**
> >
> > > 3. The prediction results are not reported with standard deviation, which makes it hard to tell whether the performance is statistically significant.
> >
> > We agree with this valuable comment. However, it is important to note that the currently available test data for ddG prediction is limited, and the performance of the methods varies significantly across different protein-protein interactions (see Table 6 in the paper). As a result, the traditional bootstrapping approach [6, 7] or estimating the standard deviation across PPIs are not useful due to the high variance. This may explain why statistical significance on SKEMPI predictions has  been evaluated before (to the best of our knowledge). In fact, none of the methods we found and mention in the "Related Work" section assesses statistical significance. Nonetheless, we have now evaluated the stability of our method by retraining our model using two additional random seeds and reporting the mean and standard deviation of the test results. We provide the updated table 2 from the main paper below:
> >
> > | Category                                  | Method                   | Spearman $\uparrow$ | Pearson $\uparrow$ | Precision $\uparrow$ | Recall $\uparrow$ | ROC AUC $\uparrow$ | MAE $\downarrow$ | RMSE $\downarrow$ |
> > |-------------------------------------------|--------------------------|----------------------|--------------------|-----------------------|-------------------|---------------------|-------------------|---------------------|
> > | Force field simulations                   | Flex ddG        | $0.54$               | $0.57$             | $0.63$                | $0.62$             | $0.84$              | $1.60$            | $2.00$              |
> > |                                           |                          |                      |                    |                       |                   |                     |                   |                     |
> > | Machine learning                          | GEMME           | $\underline{0.38}$     | 0.41               | $\textbf{0.60}$         | 0.49              | 0.74                | 2.16              | 2.81                |
> > | Machine learning | MSA Transformer | $0.37$               | $\underline{0.45}$   | $0.51$                | $0.38$            | $\underline{0.76}$   | $5.99$            | $6.77$              |
> > | Machine learning | ESM-IF          | $0.32$               | $0.31$             | $0.36$                | $0.28$            | $0.69$              | $1.84$            | $2.11$              |
> > | Machine learning | RDE-Net.        | $0.24$               | $0.30$             | $\underline{0.54}$      | $\textbf{0.65}$     | $0.67$              | $\underline{1.70}$  | $\underline{2.02}$    |
> > | Machine learning | PPIformer (ours)| $\textbf{0.44} \pm 0.023$ | $\textbf{0.46} \pm 0.029$ | $\textbf{0.60} \pm 0.014$ | $\underline{0.61} \pm 0.012$ | $\textbf{0.78} \pm 0.018$ | $\textbf{1.64} \pm 0.011$ | $\textbf{1.94} \pm 0.005$ |
> >
> > The test performance for the individual training runs is provided below. Seed #1 corresponds to the originally reported performance of PPIformer.
> >
> > | Method                   | Spearman $\uparrow$ | Pearson $\uparrow$ | Precision $\uparrow$ | Recall $\uparrow$ | ROC AUC $\uparrow$ | MAE $\downarrow$ | RMSE $\downarrow$ |
> > |--------------------------|----------------------|--------------------|-----------------------|-------------------|---------------------|-------------------|---------------------|
> > | PPIformer (ours)          | $0.44 \pm 0.023$               | $0.46 \pm 0.029$             | $0.60 \pm 0.014$                | $0.61 \pm 0.012$            | $0.78 \pm 0.018$              | $1.64 \pm 0.011$            | $1.94 \pm 0.005$              |
> > | PPIformer (ours, seed #1)          | $0.42$               | $0.46$             | $0.58$                | $0.61$            | $0.77$              | $1.64$            | $1.94$              |
> > | PPIformer (ours, seed #2)          | $0.47$               | $0.49$             | $0.60$                | $0.62$            | $0.80$              | $1.62$            | $1.94$              |
> > | PPIformer (ours, seed #3)          | $0.42$               | $0.44$             | $0.61$                | $0.60$            | $0.77$              | $1.64$            | $1.94$              |

---

> > > ### Author Response · Authors · 2023-11-21
> > > **Official Comment by Authors (Part 3)**
> > >
> > > **Questions**
> > >
> > > > 1. Would it be possible to construct the message passing between interfaces with GVP, another MPNN model that aggregates with protein scalar and vector features?
> > >
> > > Yes, it is technically possible to construct the encoder f using GVP (or more precisely the GVP-GNN message passing network based on GVPs) [3] instead of Equiformer [4]. Geometric graph neural networks, i.e. those capable of learning from scalar, vector and higher-degree features, are an active research field providing a broad range of architectures [5]. GVP-GNN is a pioneering work in the field, and Equiformer is one of the most recent developments. While there is no direct comparison between Equiformer and GVP-GNN, the Equiformer architecture was shown to outperform E(n)-GNN [10] and PaiNN [11], approaches similar to GVP-GNN (please see [5] for a justification of the similarly of GVP-GNN with E(n)-GNN and PaiNN). Compared to GVP-GNN, Equiformer employs an attention mechanism and updates vector features based on previous vector features, rather than solely from scalars at each layer. We will cite and discuss GVP-GNN in the final version of the paper.
> > >
> > > > 2. Why use log odds ratio for predicting ddG, instead of training a regressor to directly predicting the ddG values?
> > >
> > > We chose to use log odds ratios due to the thermodynamic motivation discussed in Section 4.4. In particular, log odds ratio offers two important advantages: (i) it introduces an inductive bias of antisymmetry (i.e. $\Delta \Delta G_{mut \rightarrow wt} = - \Delta \Delta G_{wt \rightarrow mut}$), and (ii) it enables seamless fine-tuning after log-likelihood based pre-training, without introducing any additional parameters. Point (ii) allows us to fully unlock the potential of the self-supervised pre-training. Specifically, employing log odds for $\Delta \Delta G$ prediction, the self-supervised objective can be interpreted as learning to estimate $\Delta G$ (see Section 4.4).
> > >
> > > We validate our argument by performing two additional ablation experiments. First, following [8] and [9], we implement a baseline regressor that performs two forward passes to predict $\Delta \Delta G$: one for the wild-type structure and another for a mutated structure (i.e. with replaced  $F_0$ features capturing one-hot encoded amino acid types). Then, it estimates $\Delta \Delta G$ by taking the average of node embeddings corresponding to residues being mutated in both structures, concatenating them, and applying a simple two-layer regressor of the form: $$
> > > Linear(256, 128) \rightarrow ReLU \rightarrow Linear(128, 1).
> > > $$ This approach results in a significant drop in performance (see the first row “Naive regressor” in the table below). Second, we improve this naive design by incorporating antisymmetry: removing biases in the Linear layers, replacing ReLU with Tanh (an odd non-linear function), and subtracting averaged embeddings instead of concatenating them. This improvement leads to a performance boost compared to the naive baseline, but still strongly underperforms the log odds ratio in terms of practically important metrics (see the second row “Antisymmetric regressor” in the table below).
> > >
> > > | Method                   | Spearman $\uparrow$ | Pearson $\uparrow$ | Precision $\uparrow$ | Recall $\uparrow$ | ROC AUC $\uparrow$ | MAE $\downarrow$ | RMSE $\downarrow$ |
> > > |--------------------------|----------------------|--------------------|-----------------------|-------------------|---------------------|-------------------|---------------------|
> > > | PPIformer (Naive regressor)          | $0.298$               | $0.330$             | $0.204$                | $0.258$            | $0.724$              | $1.844$            | $2.210$              |
> > > | PPIformer (Anysymmetric regressor)          | $0.348$               | $0.382$             | $0.494$                | $0.626$            | $0.658$              | $1.720$            | $1.964$              |
> > > | PPIformer (ours)          | $0.44 \pm 0.023$               | $0.46 \pm 0.029$             | $0.60 \pm 0.014$                | $0.61 \pm 0.012$            | $0.78 \pm 0.018$              | $1.64 \pm 0.011$            | $1.94 \pm 0.005$

---

> > > > ### Author Response · Authors · 2023-11-21
> > > > **Official Comment by Authors (Part 4)**
> > > >
> > > > **References**
> > > >
> > > > [1] Luo, Shitong, et al. "Rotamer Density Estimator is an Unsupervised Learner of the Effect of Mutations on Protein-Protein Interaction." bioRxiv (2023): 2023-02.
> > > >
> > > > [2] Shan, Sisi, et al. "Deep learning guided optimization of human antibody against SARS-CoV-2 variants with broad neutralization." Proceedings of the National Academy of Sciences 119.11 (2022): e2122954119.
> > > >
> > > > [3] Jing, Bowen, et al. "Learning from protein structure with geometric vector perceptrons." arXiv preprint arXiv:2009.01411 (2020).
> > > >
> > > > [4] Liao, Yi-Lun, et al. "EquiformerV2: Improved Equivariant Transformer for Scaling to Higher-Degree Representations." arXiv preprint arXiv:2306.12059 (2023).
> > > >
> > > > [5] Joshi, Chaitanya K., et al. "On the expressive power of geometric graph neural networks." arXiv preprint arXiv:2301.09308 (2023).
> > > >
> > > > [6] Jumper, John, et al. "Highly accurate protein structure prediction with AlphaFold." Nature 596.7873 (2021): 583-589.
> > > >
> > > > [7] Notin, Pascal, et al. "Tranception: protein fitness prediction with autoregressive transformers and inference-time retrieval." International Conference on Machine Learning. PMLR, 2022.
> > > >
> > > > [8] Liu, Xianggen, et al. "Deep geometric representations for modeling effects of mutations on protein-protein binding affinity." PLoS computational biology 17.8 (2021): e1009284.
> > > >
> > > > [9] Shan, Sisi, et al. "Deep learning guided optimization of human antibody against SARS-CoV-2 variants with broad neutralization." Proceedings of the National Academy of Sciences 119.11 (2022): e2122954119.
> > > >
> > > > [10] Satorras, Vıctor Garcia, Emiel Hoogeboom, and Max Welling. "E (n) equivariant graph neural networks." International conference on machine learning. PMLR, 2021.
> > > >
> > > > [11] Schütt, Kristof, Oliver Unke, and Michael Gastegger. "Equivariant message passing for the prediction of tensorial properties and molecular spectra." International Conference on Machine Learning. PMLR, 2021.
> > > >
> > > > [12] Laroche, Yves, et al. "Recombinant staphylokinase variants with reduced antigenicity due to elimination of B-lymphocyte epitopes." Blood, The Journal of the American Society of Hematology 96.4 (2000): 1425-1432.

---

> ### Comment · Reviewer_hYdu · 2023-11-22
>
> Thanks for updating these extensive experimental results and detailed explanations in such a small time window. The majority of my concerns are well-addressed. While the proposed method does not always achieve top performance (which is usually expected for a conference paper to be accepted), I reckon many investigations and analyses are valuable. I have thus raised my score to 6.
>
> Also, since the authors have made great efforts during the rebuttal, it would be nice if these investigations could be integrated into the final version (if it gets accepted).
>
> Reminder: Please revise your paper to fit into 9 pages as required.

---

### Comment · Reviewer_Sjjh · 2023-11-21
**A Kind Reminder to Authors**

Hi, authors of PPIFormer,

Since the deadline of discussion period is approaching, I want to know whether you are going to reply to our reviewers' quesitons. I understand that most of us gave a relatively high score, but this does not mean that we have no doubt about the details of PPIformer.

To be specific, I sincerely hope you can answer the following questions:
(1) Can you please upload the code to some anomynous repo? **I have reproduced the method described in your paper but the performance is far away from your reported one**. Therefore, I would be pleased to know how you implement this algorithm (e.g., the use of Equiformer, the masked amino acid modeling, fine-tuning the pretrained model)?

(2) Can you please report the performance of PPIformer on the same split of RDE-Net in Skempi v2? As said by Reviewer hYdu, we may suspect that you evaluated models on a cherry-picked test set. The number of test set is so small, and we want to see more comprehensive evaluations.

Due to this reason, I regretfully inform you that I adjusted my score from 6 to 3. But I would be happy to raise it if my concerns are addressed.

Thanks.

---

> ### Author Response · Authors · 2023-11-21
>
> We thank the reviewer very much for the high attention dedicated to the submission. We would like to sincerely apologize for the delay with answering the reviews, we are trying our best to run as many experiments as possible to address all the suggestions proposed by the 5 reviewers. We plan on submitting our official answer to all the reviews today.

---

> > ### Author Response · Authors · 2023-11-21
> >
> > We understand the reviwer's concerns, and are doing our best to clarify all the raised points by all the five reviewers in the limited time of rebuttal.
> >
> > > (1) Can you please upload the code to some anomynous repo? I have reproduced the method described in your paper but the performance is far away from your reported one. Therefore, I would be pleased to know how you implement this algorithm (e.g., the use of Equiformer, the masked amino acid modeling, fine-tuning the pretrained model)?
> >
> > Thank you very much for the interest in the method. We are now working to provide the anonymized version of the pre-trained models, their source code and the inference script. This should enable easy reproduction of the test results. We plan to put it on the suggested anonymized website later today or tomorrow.
> >
> > Providing easily usable training code on such a short notice is more challenging as it also requires the entire new PPIRef dataset, which is one of the key ingredients of the method (please see Figure 4 in the paper) and the current codebase is tied to the computing infrastructure. However, we plan to publish all the code, models and data after the review process and set-up a webserver for others to easily try the method.
> >
> > > (2) Can you please report the performance of PPIformer on the same split of RDE-Net in Skempi v2? As said by Reviewer hYdu, we may suspect that you evaluated models on a cherry-picked test set. The number of test set is so small, and we want to see more comprehensive evaluations.
> >
> > [Our reponse (see below)](https://openreview.net/forum?id=xcMmebCT7s&noteId=bfXV4RNeg5) now includes the performance of PPIformer on the RDE-Net splits of SKEMPI v2.0. Without introducing any changes to our architecture or hyper-parameters, PPIformer achieves competitive performance, especially on the challenging multiple-mutation subset.
> >
> > [Our response to the reviewer hYdu](https://openreview.net/forum?id=xcMmebCT7s&noteId=0tshmegDww), as well as the updated version of Appenix B, now describes the construction of our three mutually-independent test sets from three different sources. We hope it clarifies that our results are not cherry-picked from a single dataset.

---

### Author Response · Authors · 2023-11-21

We thank the reviewers for their valuable comments, which will allow us to improve the quality of the paper.

We tried our best to respond to all comments from all five reviewers including running several additional experiments.  However, responding to all 5 reviewers took significant amount of time and we sincerely apologize for the delay. We are happy to take any additional comments in the discussion.

We have also uploaded an updated version of the pdf, which mainly includes an updated Appendix B.2 “Test Datasets” that describes in detail the construction of the test sets together with few other minor edits (highlighted in blue). In order to get the response out quickly now, we have not yet included in the updated pdf of the submission all the new results presented in the responses below. However, we will include all the responses and new results in the camera-ready version, should the paper be accepted. We also plan to release the entire codebase, all the data and trained models after we incorporate all the comments from the review process.

---

### Author Response · Authors · 2023-11-22
**Our code on Anonymous Github**

We highly appreciate the new comments by the reviewers. As suggested by the reviewer Sjjh, we have prepared a demo version of our complete codebase on Anonymous Github. This includes our three Python packages [PPIformer](https://anonymous.4open.science/r/PPIformer-ICLR2024-rebuttal-912F/README.md), [PPIRef](https://anonymous.4open.science/r/PPIRef-ICLR2024-rebuttal-ABE2/README.md) and auxiliary [mutils](https://anonymous.4open.science/r/mutils-ICLR2024-rebuttal--5A88/README.md), required for easy reproduction of our test results and complete specification of our algorithms. In case the reviewers are interested in reproducing our test results, please clone all three repositories, download the weights from [Zenodo](https://zenodo.org/records/10183718) into `PPIformer-ICLR2024-rebuttal/weights/ddg_regression` and run two commands speicifed in the README file of the PPIformer repository.

---

### Meta-Review · Area_Chair_eWaQ · 2023-12-05

**Metareview:**

This work introduces PPIformer, a novel SE(3)-equivariant machine learning model, trained on PPIRef—a non-redundant database of 3D protein-protein interactions curated to reduce redundancy in datasets and improve data quality. PPIformer is pre-trained on PPIRef and fine-tuned to predict the effects of mutations on protein-protein interactions. The approach has competitive performance to other methods and generalizes to practical applications, including optimizing a human antibody against SARS-CoV-2 and enhancing staphylokinase thrombolytic activity.

**Justification For Why Not Higher Score:**

The achieved performance of the model is similar to recent strong baselines (RDE). There is no theory that supports the claims.

**Justification For Why Not Lower Score:**

The work focuses on an important open problem achieving competitive results. The introduced dataset of PPI can be useful to the community. Ablations show the importance of the different components.

---

### Decision · Program_Chairs · 2024-01-16

Accept (poster)